# Drivers of global mangrove loss and gain in social-ecological systems

Valerie Hagger [1] ✉, Thomas A. Worthington [2], Catherine E. Lovelock [1], Maria Fernanda Adame [3], Tatsuya Amano [1], Benjamin M. Brown[4], Daniel A. Friess[5,6], Emily Landis[7], Peter J. Mumby [1], Tiffany H. Morrison [8], Katherine R. O'Brien [9], Kerrie A. Wilson[10], Chris Zganjar[7] & Megan I. Saunders[11]

Mangrove forests store high amounts of carbon, protect communities from storms, and support fisheries. Mangroves exist in complex social-ecological systems, hence identifying socioeconomic conditions associated with decreasing losses and increasing gains remains challenging albeit important. The impact of national governance and conservation policies on mangrove conservation at the landscape-scale has not been assessed to date, nor have the interactions with local economic pressures and biophysical drivers. Here, we assess the relationship between socioeconomic and biophysical variables and mangrove change across coastal geomorphic units worldwide from 1996 to 2016. Globally, we find that drivers of loss can also be drivers of gain, and that drivers have changed over 20 years. The association with economic growth appears to have reversed, shifting from negatively impacting mangroves in the first decade to enabling mangrove expansion in the second decade. Importantly, we find that community forestry is promoting mangrove expansion, whereas conversion to agriculture and aquaculture, often occurring in protected areas, results in high loss. Sustainable development, community forestry, and co-management of protected areas are promising strategies to reverse mangrove losses, increasing the capacity of mangroves to support human-livelihoods and combat climate change.

Coastal ecosystems upon which people depend[1] have suffered significant declines from human impacts[2] and are being exacerbated by global climate change[3]. Ecosystem-based mitigation and adaptation strategies that rebuild marine life and deliver ecosystem services are urgently needed[4]. Mangrove forests occur in 105 countries across 5 continents[5]. Globally they enhance fisheries sustaining 4.1 million small-scale fishers[6,7], protect coasts providing flood protection worth $US 65 billion per year[8,9], and mitigate climate change storing 8.5 gigatons of carbon[10,11]. Global mangrove cover was coarsely estimated to have declined by 35% by the end of the 1990s[12]. Improvements in remote sensing have monitored a further 2.1% (3363 km²) decline between 2000 and 2016, primarily from conversion to aquaculture and agriculture[13].

[1]School of Biological Sciences, The University of Queensland, Brisbane, QLD, Australia. [2]Conservation Science Group, Department of Zoology, University of Cambridge, Cambridge CB2 3QZ, UK. [3]Australian Rivers Institute, Centre for Marine and Coastal Research, Griffith University, Brisbane, QLD, Australia. [4]Research Institute for Environment & Livelihoods, Charles Darwin University, Darwin, NT, Australia. [5]Department of Geography, National University of Singapore, Singapore, Republic of Singapore. [6]Centre for Nature-based Climate Solutions, National University of Singapore, Singapore, Republic of Singapore. [7]The Nature Conservancy, Arlington, VA, USA. [8]Australian Research Council Centre of Excellence for Coral Reef Studies, James Cook University, Townsville, QLD, Australia. [9]School of Chemical Engineering, The University of Queensland, Brisbane, QLD, Australia. [10]Queensland University of Technology, Brisbane, QLD, Australia. [11]Coasts and Ocean Research Program, Oceans and Atmosphere, Commonwealth Scientific and Industrial Research Organisation, St Lucia, QLD, Australia. ✉e-mail: v.hagger@uq.edu.au

Resources-strapped governments and conservation practitioners need to know the types of interventions that can reverse declines, and this requires understanding the drivers that (1) decrease rates of loss and (2) increase rates of gain. Previous studies have identified a suite of socioeconomic factors which enable conservation interventions, including effective management of protected areas, strong governance, community dependence on resources[14,15], control of corruption[16], and higher levels of democracy[17]. For mangroves, countries with effective regulations have mediated the pressure of high population density on mangrove losses, and offer greater protection of mangroves outside of protected areas[18]. However, recent increases in the biophysical drivers of global mangrove losses, such as shoreline erosion and extreme weather events, present additional challenges for mangrove conservation[13]. Identifying these drivers helps to allocate resources to regions where conservation may be more effective and can identify policies and programmes which can be actively implemented to improve conservation. For terrestrial forests, community forestry[19], recognition of Indigenous land tenure rights[20], and climate policy programmes[21] have led to reduced deforestation rates. In Myanmar and Kenya, community forestry is being encouraged in mangroves to improve livelihoods and mangrove conservation[22,23], however, the effects of such initiatives on mangrove forests on a global-scale are unknown and represent a major gap in our understanding of how we can reverse mangrove losses. The latter is required to meet the post-2020 global biodiversity framework of the Convention on Biological Diversity and countries' climate commitments under the Paris Agreement.

Here, we used hierarchical modelling to explore the relationship between four measures of mangrove cover change and socioeconomic and biophysical variables, accounting for national- and landscape-level variability. We used a high resolution, global time-series on mangrove cover from 1996 to 2016[5] to calculate the percentage of mangrove cover loss and gain across landscape mangrove geomorphic units[24], over 1996–2007 and 2007–2016; assessing two decades allowed us to account for recent reductions in mangrove loss[25], within the available timesteps of the Global Mangrove Watch dataset. Mangrove geomorphic units were delineated by the maximal extent of mangrove cover across 1996–2016 and classified into typologies based on geomorphic settings (see "Methods"). We assessed percent net loss, percent net gain, percent gross loss, and percent gross gain to consider differences in processes that contribute to losses that "are offset" and "not offset" by gains and vice versa. At the national scale, we assessed the effect of conservation policies and activities, including support for community forestry (measured as the number of community or social forestry case studies in countries and whether community forestry was within mangroves), restoration effort (measured as the number of mangrove restoration projects recorded per country), commitment to mangroves in Nationally Determined Contributions (NDCs), proportion of Indigenous people's land, and extent of Ramsar wetland area. These were complemented by national indicators of governance, including the level of democracy ranging from autocratic to democratic regimes, and economic complexity representing the diversified capability of a country's economy[26]. At the finer landscape scale, we assessed how indicators of economic growth and access to markets, measured as night-time lights growth[9] and travel time to nearest city[27] across mangrove geomorphic units, affected mangrove cover change. We controlled for biophysical variables known to influence mangrove extent by accounting for mangrove fragmentation[18], potential sediment availability from rivers due to anthropogenic barriers, tidal amplitude, antecedent sea-level rise (SLR) from 1993 to 2015[3], severity and frequency of droughts and tropical storms[13], extreme low temperatures[28], and coastal geomorphic type[24] also measured across mangrove geomorphic units (Fig. 1).

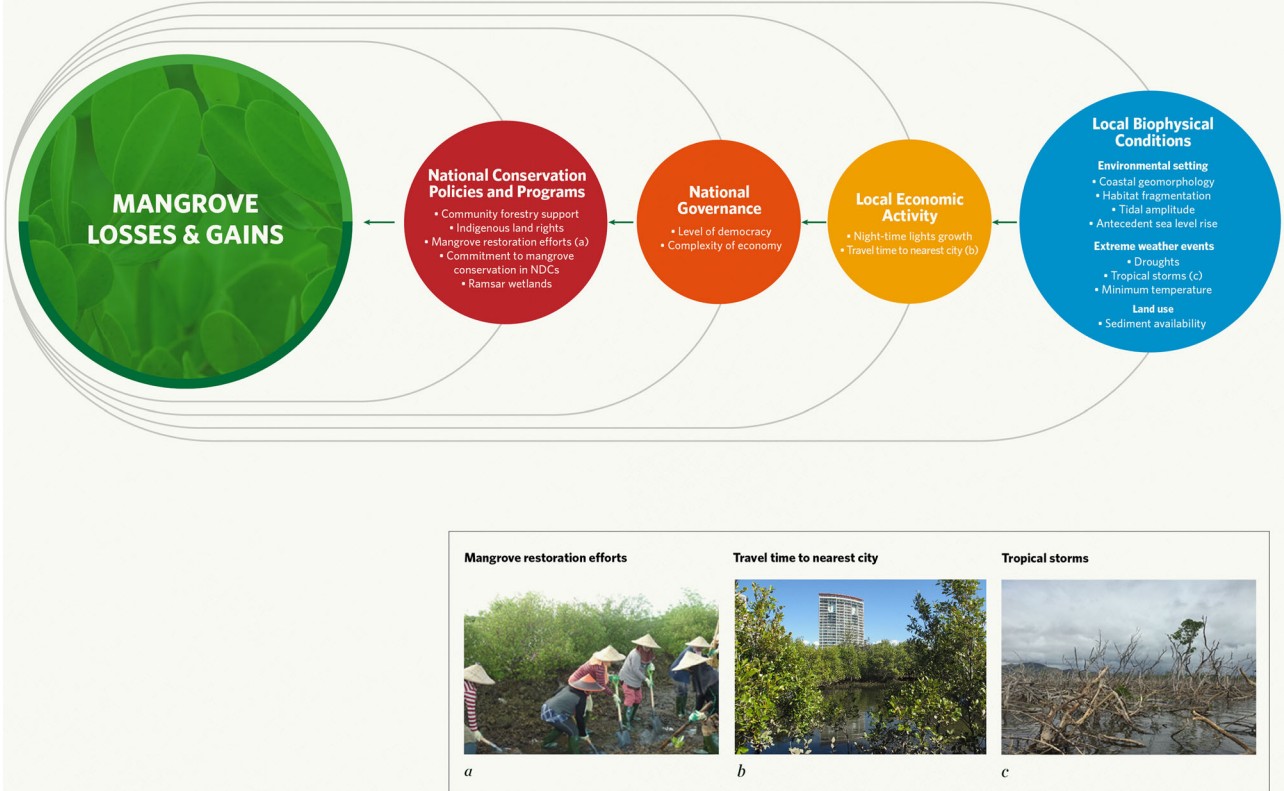

**Fig. 1 | Factors that influence global mangrove cover losses and gains (green).** Factors in our analyses were local biophysical conditions (blue) and socioeconomic variables (yellow) and national socioeconomic variables (orange and red). Examples of mangrove gain can be due to restoration of mangroves by tidal channel recreation in South Sulawesi, Indonesia (**a**), whereas loss can be due to development pressure close to cities in Gold Coast, Australia (**b**) and mortality of mangroves by a tropical storm in Yacuna, Fiji (**c**). Image credits: Rio Ahmad (**a**), Valerie Hagger (**b**), Clint Cameron (**c**).

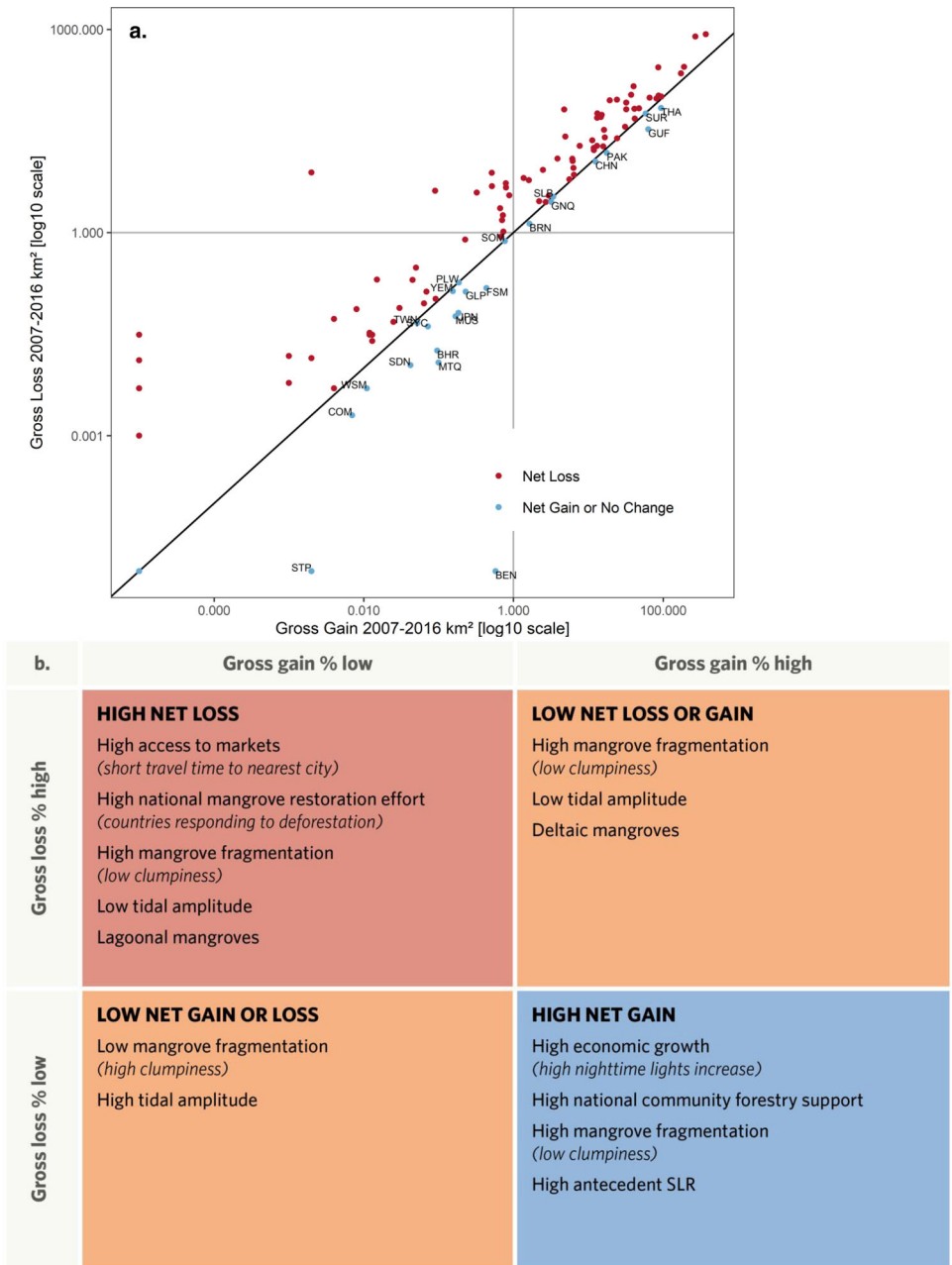

**Fig. 2 | Nexus of mangrove losses and gains. a** National gross losses and gains in mangrove cover detected within 109 mangrove countries between 2007 and 2016. Points are coloured by the country's net change that fall above or below the 1:1 line, respectively. The horizontal and vertical lines visualise where countries fall on the gradient of gross loss and gain, respectively. Countries with net gain are labelled by country code. BEN Benin, BHR Bahrain, BRN Brunei, CHN China, COM Comoros, FSM Federated States of Micronesia, GLP Guadeloupe, GNQ Equatorial Guinea, GUF French Guiana, JPN Japan, MUS Mauritius, MTQ Martinique, PAK Pakistan, PLW Palau, SDN Sudan, SLB Solomon Islands, SOM Somalia, STP Sao Tome and Principe, SUR Suriname, SYC Seychelles, THA Thailand, TWN Taiwan, YEM Yemen, WSM Samoa. **b** Socioeconomic and biophysical drivers of gradients of mangrove losses and gains between 2007 and 2016 (significant effects with a $p < 0.05$ from two sample t-tests detected in the mixed-effects models). SLR sea-level rise.

## Results and discussion

### Drivers of mangrove losses and gains

The rate of net mangrove loss declined globally from 2.74% in 1996–2007 to 1.58% in 2007–2016, consistent with previous studies[29,30]. While mangrove extent globally continued to decline, low gross losses and high gross gains led to net gains or no change in some countries (Fig. 2a). In the first decade, at the landscape scale, net mangrove losses across geomorphic units were similarly high in all geomorphic types (delta, estuary, lagoon, and open coast), and were significantly negatively associated with travel time to nearest city, clumpiness (with low clumpiness indicating high mangrove fragmentation), tidal amplitude, Standardised Precipitation-Evapotranspiration Index (SPEI, with low SPEI indicating high drought severity), and minimum temperature, and positively associated with antecedent SLR. At the national scale, net losses were positively associated with mangrove restoration effort (Fig. 3a, $R^2m/c$ [marginal/conditional] = 0.17/NA, $df = 27$, $n = 2004$). In the second decade, at the landscape scale, net losses were relatively high in lagoons and were significantly negatively associated with travel time to nearest city, clumpiness, and tidal amplitude. At the national scale, net losses were positively associated with mangrove restoration effort (Fig. 3b, $R^2m/c = 0.07/0.13$, $df = 27$, $n = 1914$).

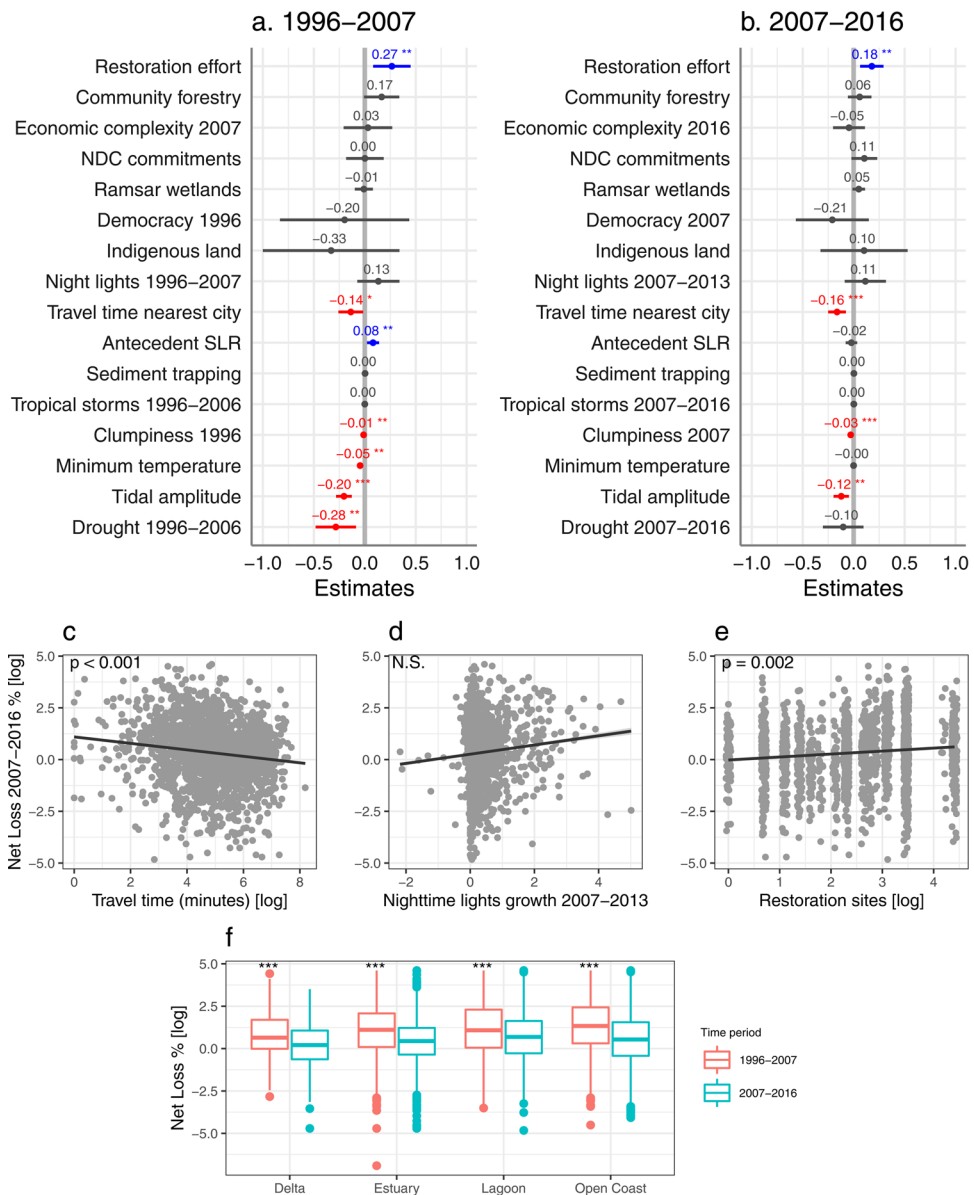

**Fig. 3 | Global drivers of mangrove net loss.** Fixed effects of socioeconomic and biophysical variables on percent net loss of mangrove cover in 1996–2007 (**a**; $n$ = 2004 mangrove geomorphic units across 53 countries) and 2007–2016 (**b**; $n$ = 1914 mangrove geomorphic units across 56 countries) showing the estimate and 95% confidence intervals from the mixed-effects models. Positive values (in blue) indicate that mangrove loss increased with an increase in the variable, while negative values (in red) indicate that mangrove loss decreased with an increase in the variable. NDC Nationally Determined Contribution, SLR sea-level rise. Relationship between percent net loss 2007–2016 and travel time to nearest city (**c**), night-time

lights growth (**d**), and mangrove restoration effort (**e**) from the mixed-effects model ($n$ = 1914), showing the fitted regression line and 95% confidence intervals. N.S. not significant. **f** Percent net loss across mangrove geomorphic types in 1996–2007 ($n$ = 2004) and 2007–2016 ($n$ = 1914) showing the median, 25th and 75th percentiles at the lower and upper hinges, and whiskers extending to the smallest and largest values at most 1.5 × inter-quartile range. $P$-values derived from two sample t-tests. Asterisks indicate significant effects at $p < 0.05$*, $p < 0.01$**, and $p < 0.001$***. Corresponding $t$-values and $p$-values for each variable are provided in Supplementary Table 10. See Fig. 5 for comparison to percent gross loss results.

Net gains of mangroves occurred in 728 geomorphic units in 64 countries in 1996–2007, which increased to 1138 units in 80 countries in 2007–2016. In the first decade, at the landscape scale, net mangrove gains were strongly negatively correlated with clumpiness, antecedent SLR, and minimum temperature. At the national scale, net gains were positively associated with mangrove restoration effort, but negatively associated with proportion of Indigenous people's land (Fig. 4a, $R^2m/c$ = 0.45/NA, $df$ = 27, $n$ = 451). In the second decade, net gains were overall strongly positively correlated with night-time lights growth and antecedent SLR at the landscape scale and community forestry effort at the national scale (Fig. 4b, $R^2m/c$ = 0.13/0.26, $df$ = 27, $n$ = 743). When assessing gross as opposed to net losses and gains, additional variables emerged as significant,

highlighting the importance of considering multiple metrics of mangrove change to give a comprehensive picture of the drivers (Fig. 2b). For gross losses, these included positive relationships with night-time lights growth, community forestry, and sediment trapping in the first decade (Fig. 5a, $R^2m/c$ = 0.16/0.35, $df$ = 27, $n$ = 2425). In the second decade, these included higher gross losses in deltas and positive relationships with NDC commitment and sediment trapping (Fig. 5b, $R^2m/c$ = 0.08/0.28, $df$ = 27, $n$ = 2637). For gross gains, these included negative relationships with democracy in the first decade (Fig. 6a, $R^2m/c$ = 0.16/0.35, $df$ = 27, $n$ = 2425) and with tidal amplitude and minimum temperature in the second decade (Fig. 6b, $R^2m/c$ = 0.14/0.32, $df$ = 27, $n$ = 2554; Supplementary Table 10 summarises significant results).

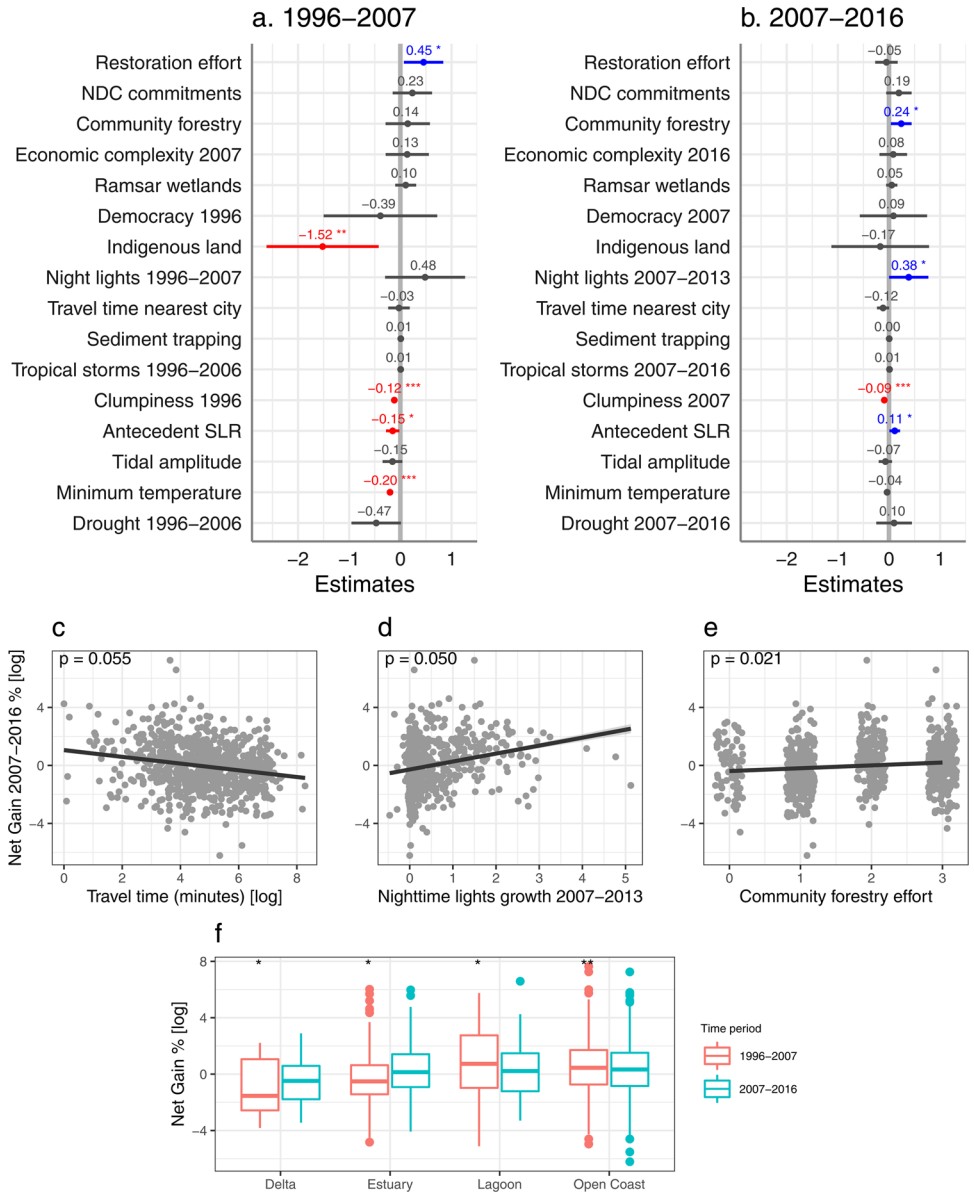

**Fig. 4 | Global drivers of mangrove net gain.** Fixed effects of socioeconomic and biophysical variables on percent net gain 1996–2007 (**a**; *n* = 451 mangrove geomorphic units across 41 countries) and 2007–2016 (**b**; *n* = 743 mangrove geomorphic units across 49 countries) showing the estimate and 95% confidence intervals from the mixed-effects models. Positive values (in blue) indicate that mangrove gain increased with an increase in the variable, while negative values (in red) indicate that mangrove gain decreased with an increase in the variable. NDC Nationally Determined Contribution. SLR sea-level rise. Relationship between percent net gain 2007–2016 and travel time to nearest city (**c**), night-time lights growth (**d**), and community forestry effort (**e**) from the mixed-effects model (*n* = 743), showing the fitted regression line and 95% confidence intervals. **f** Percent net gain across mangrove geomorphic types in 1996–2007 (*n* = 451) and 2007–2016 (*n* = 743) showing the median, 25th and 75th percentiles at the lower and upper hinges, and whiskers extending to the smallest and largest values at most 1.5 × interquartile range. *P*-values derived from two sample t-tests. Asterisks indicate significant effects at *p* < 0.05*, *p* < 0.01**, and *p* < 0.001***. Corresponding *t*-values and *p*-values for each variable are provided in Supplementary Table 10. See Fig. 6 for comparison to percent gross gain results.

## Greater mangrove losses with higher access to markets

Our analysis shows that access to markets is a strong driver of mangrove loss, likely because of associated pressures of conversion to aquaculture and agriculture[13], with greater loss in areas with shorter travel time to the nearest city than more remote areas in both decades for net loss (*t* = −2.21, *p* = 0.03 and *t* = −3.36, *p* < 0.001, respectively, Fig. 3a, b) and the second decade for gross loss (*t* = −2.59, *p* = 0.01, Fig. 5b). Remote areas are typically better conserved because protected areas in these regions have lower levels of pressure from other land uses[31]. Negative effects of human pressures on global mangrove losses have been found to be mediated in countries with stronger governance[18]. While we found that the effect of access to markets on mangrove loss varied among countries,

negative effects were only weakly reduced in countries with high levels of economic complexity and democracy (N.S., Extended Data Fig. 4a–c). We reveal that greater access to markets was also weakly associated with increased net gains in the recent decade (*t* = −1.92, *p* = 0.06, Fig. 4b, c).

## Economic growth no longer associated with mangrove losses

A key finding is a reversal of the effect of economic growth on mangrove change. Mangrove gross loss increased significantly with higher night-time lights growth (an indicator of economic growth) in the first decade (*t* = 2.05, *p* = 0.04, Fig. 5a); however, this relationship was not apparent in the second decade (*t* = 0.10, *p* = 0.92, Fig. 5b, d). In contrast, greater net gains were observed in areas with higher

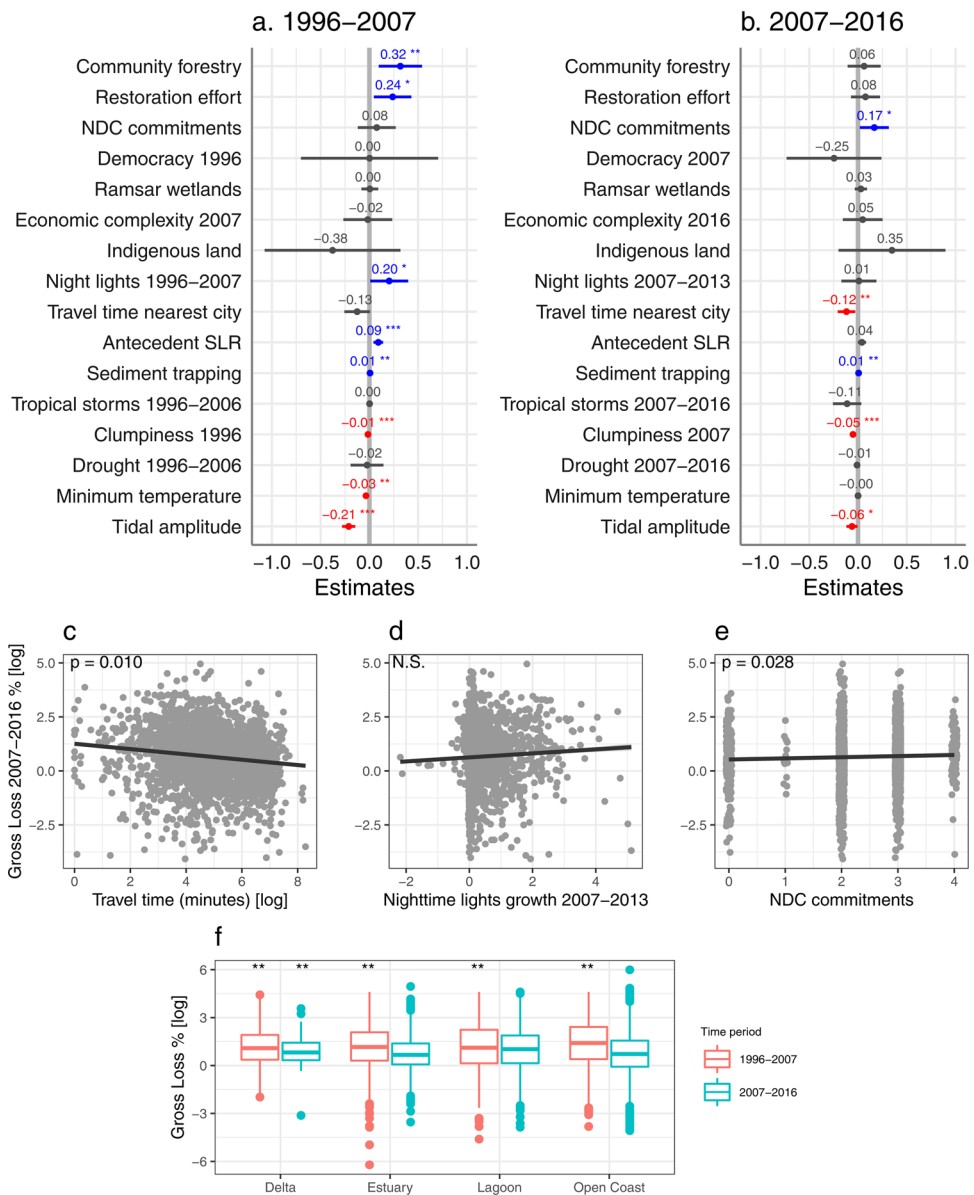

**Fig. 5 | Global drivers of mangrove gross loss.** Fixed effects of socioeconomic and biophysical variables on percent gross loss of mangrove cover in 1996–2007 (**a**; *n* = 2425 mangrove geomorphic units across 55 countries) and 2007–2016 (**b**; *n* = 2637 mangrove geomorphic units across 59 countries) showing the estimate and 95% confidence intervals from the mixed-effects models. Positive values (in blue) indicate that mangrove loss increased with an increase in the variable, while negative values (in red) indicate mangrove loss decreased with an increase in the variable. NDC Nationally Determined Contribution. SLR sea-level rise., Relationship between percent gross loss 2007–2016 and travel time to nearest city (**c**), night-time lights

growth (**d**), and NDC commitment (**e**) from the mixed-effects model (*n* = 2637), showing the fitted regression line and 95% confidence intervals. N.S. not significant. **f** Percent gross loss across mangrove geomorphic types in 1996–2007 (*n* = 2425) and 2007–2016 (*n* = 2637) showing the median, 25th and 75th percentiles at the lower and upper hinges, and whiskers extending to the smallest and largest values at most 1.5 × inter-quartile range. *P*-values derived from two sample t-tests. Asterisks indicate significant effects at $p < 0.05^*$, $p < 0.01^{**}$, and $p < 0.001^{***}$. Corresponding *t*-values and *p*-values for each variable are provided in Supplementary Table 10.

night-time lights growth in the second decade (*t* = 1.96, *p* = 0.05, Fig. 4b, d), suggesting that local communities can aid mangrove restoration or afforestation[32] or that natural regeneration occurs as sustainable development progresses. An earlier study also found that countries with higher gross national product had higher rates of mangrove loss in the 1990s[12], which coincided with an intense period of aquaculture expansion[33]. Our counterintuitive results indicate that while economic growth had a negative impact on mangrove extent in the past, recently it is correlated with mangrove expansion, potentially because of increased wealth and education, and improved agricultural productivity[34], ultimately reducing development pressure.

## Community forestry promoting mangrove gains
Our analysis reveals a strong positive association between community forestry effort and mangrove gains. Mangrove gain increased with greater community forestry effort in the second decade for both net and gross gain variables (*t* = 2.31, *p* = 0.02, Fig. 4b, e and *t* = 2.62, *p* = 0.009, Fig. 6b, e, respectively), despite higher mangrove gross losses being associated with greater community forestry effort in the first decade (*t* = 2.80, *p* = 0.005, Fig. 5a). These contradictory patterns may reflect the implementation of community forestry policies in nations responding to rapid forest deforestation in the first decade. For example, Thailand reversed from a national net loss of 6.02% of mangroves in the first decade to a net gain of 1.11% in the second

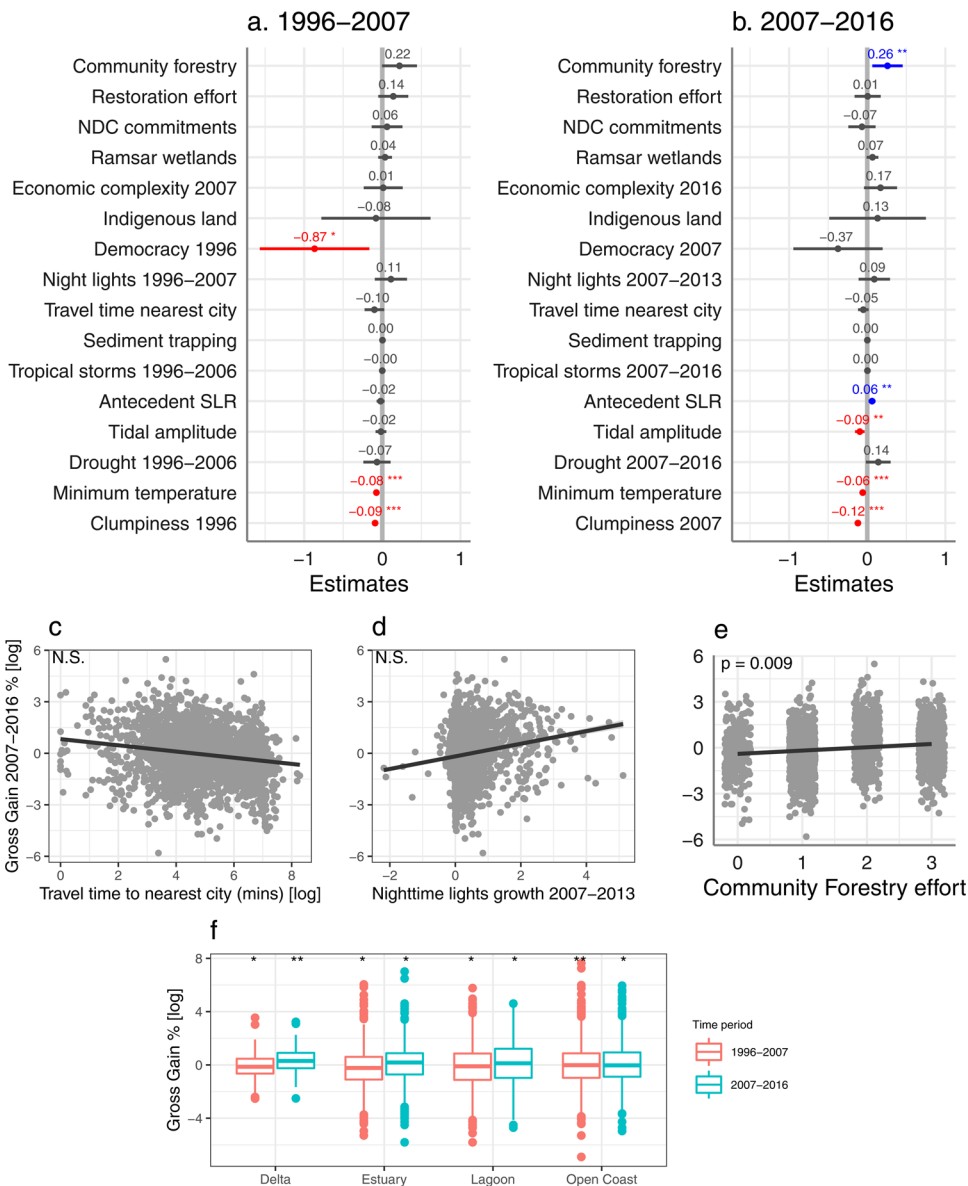

**Fig. 6 | Global drivers of mangrove gross gain.** Fixed effects of socioeconomic and biophysical variables on percent gross gain of mangrove cover in 1996–2007 (**a**; n = 2341 mangrove geomorphic units across 55 countries) and 2007–2016 (**b**; n = 2554 mangrove geomorphic units across 59 countries) showing the estimate and 95% confidence intervals from the mixed-effects models. Positive values (in blue) indicate mangrove gain increased with an increase in the variable, while negative values (in red) indicate mangrove gain decreased with an increase in the variable. NDC Nationally Determined Contribution. SLR sea-level rise. Relationship between percent gross gain 2007–2016 and travel time to nearest city (**c**), night-time lights growth (**d**), and community forestry effort (**e**) from the mixed-effects model (n = 2554), showing the fitted regression line and 95% confidence intervals. N.S. not significant. **f** Percent gross gain across mangrove geomorphic types in 1996–2007 (n = 2341) and 2007–2016 (n = 2554) showing the median, 25th and 75th percentiles at the lower and upper hinges, and whiskers extending to the smallest and largest values at most 1.5 × inter-quartile range. P-values derived from two sample t-tests. Asterisks indicate significant effects at $p < 0.05$*, $p < 0.01$**, and $p < 0.001$***. Corresponding t-values and p-values for each variable are provided in Supplementary Table 10.

decade, coinciding with extensive community forestry effort, including mangrove forestry. In the Trang province, southern Thailand, community-based mangrove management has improved mangrove conservation and local livelihoods[35]. However, there were possibly other contributing policy factors, for example, Thailand implemented a ban on mangrove charcoal production in the late 1990s[36]. Community forest management or co-management can lead to positive conservation and social outcomes, particularly in communities with de facto tenure rights and countries with low development and governance indicators[37]. For instance, titling indigenous communities has been shown to decrease clearing and disturbance in the Peruvian Amazon forests[38]. Increased national commitment to community forestry policies and programmes could potentially reverse mangrove

losses in other countries, such as Kenya and Myanmar[22,23], however, there are issues with the governance and tenure of mangroves at the land-sea interface that need resolving[39]. Even where mangroves remain owned by the state, mangrove community forestry can be successful if communities have clear policy, secure user rights, well-defined governance, and assistance from external non-government organisations[32,35].

### Mangrove restoration effort and NDC commitment likely responding to deforestation

We hypothesised that commitment to mangrove conservation and restoration in NDCs, which commenced in 2015, would point towards a stronger baseline in environmental governance and therefore enhance

mangrove conservation and that higher mangrove restoration effort would also enhance mangrove conservation. While there was overall higher mangrove net gain with higher restoration effort in the first decade ($t = 2.31$, $p = 0.021$, Fig. 4a), there was no association between national NDC commitment or restoration effort with reducing rates of loss or increasing rates of gain across mangrove geomorphic units at the end of the analysis period (2017). Instead, mangrove loss was higher in countries with higher restoration effort in both decades for net loss ($t = 2.83$, $p = 0.005$ and $t = 3.05$, $p = 0.002$, respectively, Fig. 3a, b) and the first decade for gross loss ($t = 2.44$, $p = 0.015$, Fig. 5a). Mangrove gross loss was also higher with higher NDC commitment in the second decade ($t = 2.20$, $p = 0.028$, Fig. 5b). This likely reflects the development of policies in response to deforestation, and may be further influenced by low reporting rates[40], small scale implementation[41], challenges with socio-legal aspects of restoration projects[42], and time-lags between the implementation of these policies and on-ground responses. Furthermore, despite increasing mangrove restoration efforts[43], there are still substantial losses from other factors. For example, in the Sundarbans of Bangladesh[44], mangrove gains due to large-scale mangrove afforestation were largely offset due to substantial losses, primarily from shoreline erosion[45]. Counter to other studies at regional-scales showing lower mangrove losses in selected Ramsar sites[30], the area of Ramsar wetlands per country was not positively associated with mangrove conservation at the national-scale. This is despite several countries having large to moderate mangrove cover and Ramsar extent relative to their mangrove area, including Brazil, Mexico, Australia, Mozambique, Gabon, Pakistan, China, and Iran. Many mangrove-holding countries are not party to the Ramsar convention and weaknesses in Ramsar governance occurs in urban wetlands in some countries[46].

## Biophysical drivers of mangrove losses and gains

Mangrove losses and gains were strongly dependent on landscape-scale biophysical conditions. As expected, higher mangrove fragmentation (lower clumpiness) was strongly associated with greater mangrove loss in both decades for net and gross loss variables (net loss %: $t = -2.91$, $p = 0.004$ and $t = -3.52$, $p < 0.001$, respectively, Fig. 3a, b; gross loss %: $t = -4.15$, $p < 0.001$ and $t = -8.00$, $p < 0.001$, respectively, Fig. 5a, b and Supplementary Fig. 2a, b)[18], but was also related to greater mangrove regeneration in both decades for net and gross gain variables (net gain %: $t = -5.51$, $p < 0.001$ and $t = -4.79$, $p < 0.001$, respectively, Fig. 4a, b; gross gain %: $t = -23.08$, $p < 0.001$ and $t = -14.08$, $p < 0.001$, respectively, Fig. 6a, b and Supplementary Fig. 3a, b). Increased loss was also associated with reduced sediment availability (higher sediment trapping) in both decades for gross loss ($t = 2.93$, $p = 0.003$ and $t = 2.83$, $p = 0.005$, respectively, Fig. 5a, b, Supplementary Fig. 2c, d), likely due to lower levels of sediment accretion that lead to the dominance of erosive forces on minerogenic mangrove shorelines[47]. However, sediment availability was not associated with mangrove gain, likely because the sediment trapping index did not measure longshore sediment supplies or sediment increases that could be coming from catchment deforestation and erosion. Seaward expansion of mangrove forests has been associated with tidal flat accretion from sediment transported by tidal currents and waves, offsetting large declines of fluvial sediments[48], and further facilitating sediment capture[49]. Thus, there is a data limitation at the global scale to assess sediment processes on mangrove change. Higher mangrove loss was also observed in areas with smaller tidal amplitude in both decades for net and gross loss variables (net loss %: $t = -5.17$, $p < 0.001$ and $t = -3.16$, $p = 0.002$, respectively, Fig. 3a, b; gross loss %: $t = -6.16$, $p < 0.001$ and $t = -2.11$, $p = 0.035$, respectively, Fig. 5a, b, Supplementary Fig. 2e, f), possibly because these areas were more vulnerable to low sediment supply and antecedent SLR. While there was evidence of mangrove dieback in response to severe droughts[13] and extreme low temperatures[28] in the first decade, demonstrated as higher net loss

with lower SPEI ($t = -2.80$, $p = 0.005$, Fig. 3a) and higher net and gross loss with lower minimum temperature of the coldest month ($t = -3.21$, $p = 0.001$, Fig. 3a and $t = -2.82$, $p = 0.005$, Fig. 5a, Supplementary Fig. 2i, respectively), mangrove gains were also higher in areas with lower minimum temperatures in the first decade for net gain ($t = -6.73$, $p < 0.001$, Fig. 4a) and both decades for gross gain ($t = -5.98$, $p < 0.001$ and $t = -5.49$, $p < 0.001$, respectively, Fig. 6a, b, Supplementary Fig. 3g, h), indicating potential regeneration after disturbance[50]. The decrease in the rate of mangrove loss in the second decade was apparent across all coastal geomorphic types, yet net losses remained relatively high in lagoonal mangroves[24] (N.S., Fig. 3f). This highlights a potential vulnerability that has been underreported, as previously studies have articulated concern that mangroves in deltas are under particular threat[51]. While we found gross losses to be relatively high in deltaic mangroves in the second decade ($t = 2.17$, $p = 0.030$, Fig. 5f), substantial mangrove gains have been reported in deltas in Asia and Africa due to sedimentation and shoreline accretion[52] and we found losses being offset by relatively high gross gains in deltaic mangroves in the second decade ($t = 2.89$, $p = 0.004$, Fig. 6f).

Like any large analysis of complex data there were a number of limitations to the research. We were restricted to the use of globally-available datasets for the estimation of indicators, therefore our models provide a semi-empirical indication of the socio-economic and biophysical conditions under mangrove change. Governance indices were not available for 46 mangrove countries, particularly small islands, which were therefore excluded from the modelling (Supplementary Data 1). To avoid bias because of inflated percent change in small geomorphic units, we also removed units smaller than 100 ha (see "Methods"), excluding another three countries. The resultant dataset comprised 89% and 90% of the 2016 global mangrove extent for the two decades; therefore our modelling represents drivers of the majority of mangrove holdings. Correlative models do not translate to causation, therefore we have used expert interpretation to understand factors related to significant changes.

## Hotspots of mangrove losses and gains

We examined outliers ('hotspots') to assess the influence of factors that aren't easily represented in global-scale models[15] and to assess geomorphic units that were excluded from the modelling. To do so, we identified hotspots where mangrove net and gross loss and gain in geomorphic units was significantly more or less than the country-average for the period 2007–2016 (Fig. 7a, b, see "Methods") and used expert interpretation to identify factors contributing to hotspots (Supplementary Tables 1 and 2). Several geomorphic units with high loss coincided with ecosystem conversion to aquaculture and agriculture, including areas in Indonesia, French Guiana, Panama, Pakistan, Suriname, and Thailand that are partially or fully designated as protected areas. This may be because of conflict between local people and government conservation efforts or the challenge of managing and monitoring large reserves. However, elsewhere protected areas appear to have been effective in conserving mangroves and are associated with low losses in areas in Ecuador, Guyana, and Ghana and high gains in Gambia and Venezuela. Other geomorphic units with high gains were located in relatively remote and inaccessible regions in Guinea, Mozambique, and Micronesia. Remote protected areas can still experience significant loss, for example in Amapa, Brazil where high percent loss was likely influenced by mud waves originating from the Amazon River[53].

Identifying priorities for mangrove conservation and restoration in coupled social-ecological systems[54] is necessary to ensure sustainable use of mangrove resources. Our global study confirms that high access to markets is a strong driver of mangrove deforestation with conversion to aquaculture and agriculture causing hotspots of mangrove loss, often in protected areas. Counterintuitively, high economic

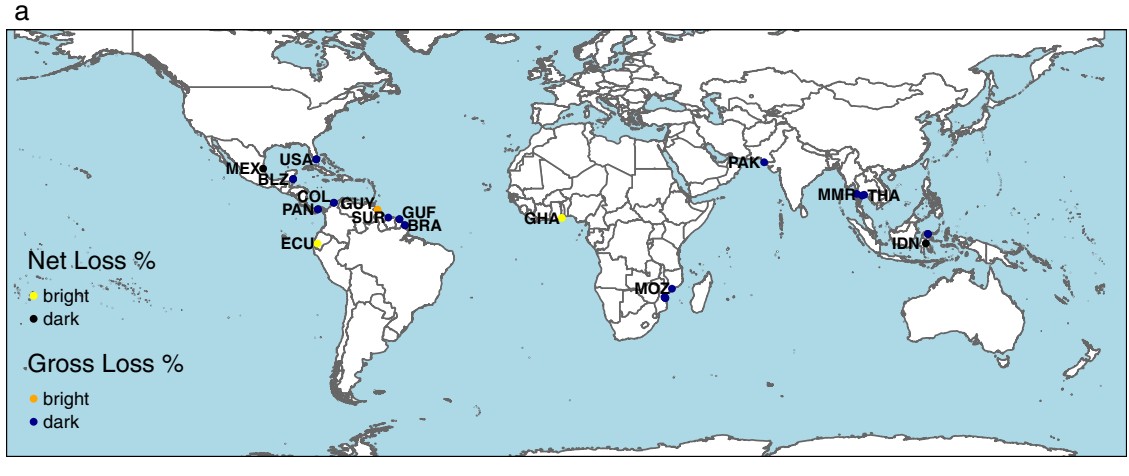

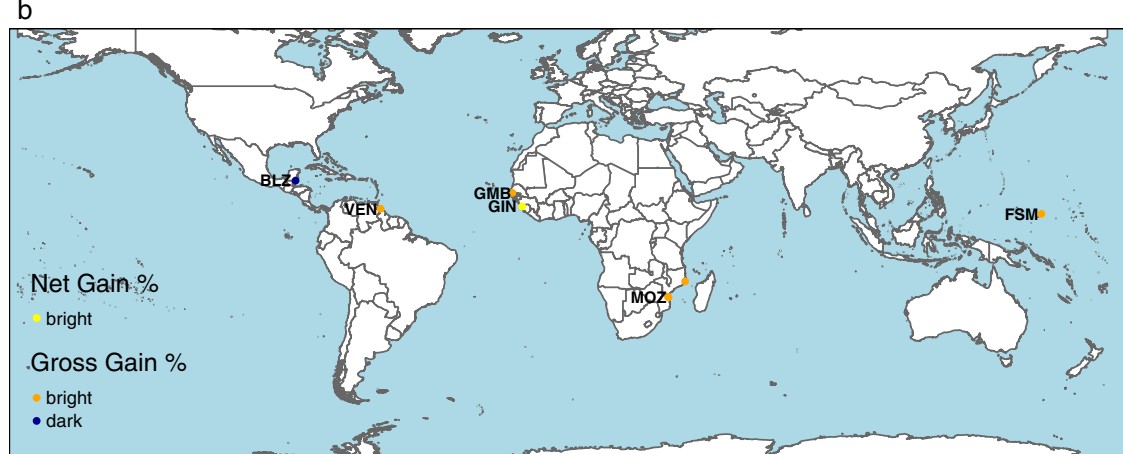

**Fig. 7 | Mangrove loss and gain hotspots.** Location of mangrove geomorphic units where percent mangrove net and gross loss in 2007–2016 is significantly (two standard deviations) less (bright) or more (dark) than the country average (**a**; *n* = 399 and 525 mangrove geomorphic units larger than 50 km² across 53 and 58 countries for net and gross loss, respectively); and percent mangrove net and gross gain in 2007–2016 is significantly more (bright) or less (dark) than the country average (**b**; *n* = 126 and 522 mangrove geomorphic units larger than 50 km² across 33 and 57 countries for net and gross gain, respectively). There are some over-lapping hotspots; seven net loss dark spots are under gross loss dark spots; one net gain bright spot is under a gross gain bright spot. Details of each hotspot and the likely reasons are listed in Supplementary Tables 1 and 2. Countries with hotspots are labelled by country code. BRA Brazil, BLZ Belize, COL Colombia, ECU Ecuador, FSM Micronesia, Federated States of, GHA Ghana, GIN Guinea, GMB Gambia, GUF French Guiana, GUY Guyana, IDN Indonesia, MEX Mexico, MOZ Mozambique, MMR Myanmar, PAN Panama, PAK Pakistan, SUR Suriname, THA Thailand, USA United States, VEN Venezuela, Bolivian Republic of. Natural Earth data was used for country boundaries [https://www.naturalearthdata.com/downloads/10m-cultural-vectors/].

growth can be compatible with mangrove expansion. Investment in community or collaborative management of mangrove forests that recognise local rights and governance[35] are promising strategies for regaining mangrove cover and enforcing mangrove protected areas, assuming that enforcement of protected areas is effective[55]. More time is required to detect positive effects from implementation of policies such as NDC commitments and restoration targets. We provide a novel approach for future assessments, spotlighting places performing well with high gain rates or low loss rates, to guide the development of effective conservation interventions. Such interventions can help end poverty, support food security, promote well-being, combat climate change, and promote a resilient ocean, and are critical to achieving multiple United Nations Sustainable Development Goals and the Decades of Ecosystem Restoration and Ocean Science for Sustainable Development.

## Methods
**Mangrove cover change variables.** We used the Global Mangrove Watch (GMW) v2.0 dataset from 1996 to 2016[56] to calculate four response variables across landscape mangrove geomorphic units[24] over two time periods, 1996–2007 and 2007–2016: (1) percent net loss

(units that had a net change in mangrove cover of <0, converted to a positive number), (2) percent net gain (units that had a net change in mangrove cover of >0), (3) percent gross loss (units that had a decrease in mangrove cover, not accounting for any increase), and (4) percent gross gain (units that had an increase in mangrove cover, not accounting for any decrease). Percent variables were calculated relative to the area at the start of the time period and were log transformed to meet the assumptions of the statistical models. We initially also considered 5 primary response variables (Supplementary Table 3), including net change in mangrove area ranging from negative (loss) to zero (no change) to positive (gain), however, the data did not meet model assumptions of equal variance (Supplementary Table 9). It was therefore necessary to separate areas of net loss and net gain and areas of gross loss and gross gain to remove zeros and log-transform to achieve normal distribution. Area of mangrove change was correlated with size of the mangrove geomorphic unit (higher area of mangrove loss or gain in bigger units), therefore we included geomorphic unit size as an explanatory variable in the models with primary response variables. We selected the transformations of these primary variables - percent net loss, percent net gain, percent gross loss, and per-cent gross gain to include in the analysis, because the percent changes

control for differences in relative sizes of geomorphic units and because net change alone can underestimate the extent of change[57].

Examining mangrove change across geomorphic settings is likely to be relevant to socioeconomic and environmental conditions. Mangroves occur in the intertidal zone in diverse coastal geomorphic settings (e.g., deltas, estuaries, lagoons) shaped by rivers, tides, and waves[58,59]. The distribution, structure, and productivity of mangroves varies spatially with regional climate and local geomorphological processes (e.g., river discharge, tidal range, hydroperiod, and wave activity) that control soil biogeochemistry[60-63]. These geomorphic settings are defined by natural landscape boundaries (e.g., catchments/bays) which also often delineate boundaries of human settlements. A global mangrove biophysical typology v2.2 dataset[64] was used for the delineation of landscape mangrove geomorphic units, which used a composite of the GMW dataset from the 1996, 2007, 2010, and 2016 timesteps to classify the maximal extent of mangrove cover into 4394 units (classified as delta, estuarine, lagoon or open coast). The mangrove geomorphic units do not include non-mangrove patches, unless they have been lost from the unit over time. The mean size of geomorphic units was 33.63 ha. Some splits of geomorphic units were undertaken to reduce size and divide by country boundaries. The four largest deltas (northern Brazil Delta ID 70000, Sundarbans Delta ID 70004, Niger Delta ID 70009, and Papua coast Delta ID 70013) were split into 4, 5, 4, and 2 units, respectively to aid with data processing. Mangrove geomorphic units that overlapped two countries (Peru/Ecuador, Singapore/Malaysia, and Papua New Guinea/ Australia) were split by the national boundary.

The country governing each geomorphic unit was assigned to match national-level variables to geomorphic units. To capture mangroves that are mapped outside of country coastline boundaries, we did a union of the GADM country shapefile v3.6[65] and the Exclusive Economic Zones (EEZs) v11[66]. The following manual country designations were made to resolve overlapping claims in the EEZs: (1) Hong Kong was merged with China as Hong Kong does not have a mapped EEZ; (2) The overlapping claim of Sudan/Egypt was maintained as a joint Sudan/Egypt designation, as this is an area of disputed land called the Halayib Triangle. However, for this study, mangrove units within this area were assigned to Egypt because Egypt currently has military control over the area; (3) Mayotte (claimed by France and Comoros) was assigned to Mayotte as it is a separate overseas territory of France recognised in GADM that has different socioeconomic variables; (4) The protected zone established under the Torres Strait Treaty was assigned to Australia as these islands are Australian territory.

Areas of mangrove cover in 1996, 2007, and 2016, and gross losses and gains in each geomorphic units over the two time periods were assessed in ArcMap 10.8[67]. Percent losses and gains were calculated in R 4.0.2[68]. In using the GMW mapping, a minimum mapping unit of 1 ha is recommended for reliable results[5], therefore we removed all geomorphic units less than 1 ha from the analysis, which reduced the available sample size from 4394 across 108 countries to 4235 units across 108 countries. In calculating percent net gains, 11 and 12 of the units returned infinity values for 1996–2007 and 2007–2016, respectively, because there was no initial mangrove cover. In these instances, 100% gain was assigned to these units.

## Socioeconomic variables (Supplementary Table 4)
**Economic growth.** Previous global analyses of mangroves have been limited by data availability on economic activity to national metrics, such as a country's Gross Domestic Product (GDP)[12,18]. Night-time lights satellite data provide local measures of economic activity that are comparable through time and available globally[9,69]. The data improve estimates of GDP in low to middle income countries[69] and are strongly correlated with local indicators of human development[70] and electricity consumption and GDP at the national-level[71]. We used the Night-time Lights Time Series v4[72] stable lights data, where transient lights

that are deemed ephemeral, e.g., fires, have been filtered out and non-lit areas set to zero[73], choosing the newer satellites where applicable[70]. As a proxy for local economic growth, we calculated the change in annual average stable lights within a 100 km buffer of the centroid of each geomorphic unit from 1996 to 2007 and 2007 to 2013 (no data available past 2013) using the 'raster' package in R[74]. The 100 km buffer was chosen to account for pressures from human activity within and surrounding the mangrove area, and to avoid bias with larger spatial units[70].

**Market accessibility.** Travel time to the nearest major market (national or provincial capital, landmark city, or major population centre) has been shown to be a stronger predictor of fish biomass on coral reefs than population density or linear distance to markets[27]. We used the global map of travel time to cities for 2015[75] to estimate the average travel time from each geomorphic unit to the nearest city via surface transport using the 'raster' package in R[74], as an indicator of access to markets to trade commodities (e.g., rice, shrimp, palm oil).

**Economic complexity.** Previous studies have examined the effect of GDP on mangrove change[18], however, this is a blunt measure of country capability. Measuring a country's economic complexity, that is the diversified capability of a nation's economy, is preferable. For example, a country with high GDP but low economic complexity can be prone to regulatory capture by high-value natural resource industries and resource corruption[26]. Therefore, we used the Economic Complexity Index (ECI)[76] for countries as an indicator of regulatory independence. The ECI had better coverage of countries in later years (Supplementary Table 4), therefore the ECI for the end of the time periods was used (2007 and 2016), although we recognise this may reduce the detection of trends because of potential time lags in impacts.

**Democracy.** We used the Varieties of Democracy (VDEM) index v10 which measures a country's degree of freedom of association, clean elections, freedom of expression, elected executives, and suffrage[77], and has been indicated to influence NDC ambition in countries to address climate change[78]. We adopted the VDEM index for the start of the time periods (1996 and 2007) to account for potential time lags in impacts.

**Community forestry support.** We determined the extent that community forestry (CF) is implemented across countries through a systematic review of articles returned in the Web of Science database (Core collection; Thomson Reuters, New York, U.S.A.). We used the search terms: TS = ("community forestry" OR "community-based forestry" OR "social forestry") AND (TI = "country" OR AB = "country") to identify how many CF case studies were reported in each country, and whether any were in mangroves. As scientific literature is biased towards particular regions, we also reviewed relevant FAO global studies[79-81] and online databases (ICCA registry[82] and REDD projects database[83]) to identify additional case studies (Supplementary Fig. 5). We then generated scores of 0–3 for each country based on summing values assessed using these criteria: +1 (1–50 CF case studies); +2 (>50 CF case studies); +1 (CF case study in mangroves). There may have been some double counting as we counted the number of case studies in each article, and we will have missed CF projects not published or communicated in English. However, this is likely to have had a limited impact on the scoring method.

**Indigenous land.** The proportion of Indigenous peoples' land versus other land per country was calculated from national-level data[84]. Whilst this study involved Indigenous peoples' land mapping at a global scale, the spatial data was not published, and thus we could only evaluate the influence of Indigenous land at the national level rather than local level.

**Restoration effort.** The number of mangrove restoration sites per country was calculated from combining two datasets collated by C. Lovelock (2020) and Y.M. Gatt and T.A. Worthington (2020) identifying mangrove restoration project locations from web searches in English and for scientific and grey literature using Google Scholar. Duplications were removed and the number of sites was used as an indicator of effort. This will underrepresent effort in countries with few, large sites, and where restoration projects are not published or communicated in English.

**Climate commitments.** The Paris Agreement is a global programme for countries to commit to climate action by submitting Nationally Determined Contributions (NDCs) to the United Nations Framework for the Convention of Climate Change (UNFCCC). First, we reviewed NDCs for mangrove-holding nations from the NDC Registry[85] submitted as of 07/01/2021 to determine the extent that mangroves or coastal ecosystems were included in national climate policy (scoring method in Supplementary Table 4). We hypothesised that countries with mangrove or coastal ecosystem NDCs may be more likely to promote mangrove conservation or restoration. While the first NDCs were submitted around 2015, at the end of our time series, we suspected higher commitments would point towards a stronger baseline in environmental governance. Most countries submitted updated or second NDCs during 2021 however these were not considered relevant to the time periods assessed. Google Translate was used to interpret NDCs in languages other than English.

**Ramsar wetlands.** The ecological character of Ramsar wetlands have been found to be significantly better than those of wetlands generally[86]. The area of Ramsar coastal and marine wetlands from the Ramsar Sites Information Service[87] was calculated per country. Thirty-eight mangrove-holding countries are not signatories to the Ramsar Convention, and these countries were assigned a value of 0. The area of Ramsar wetlands per country was scaled by dividing by the country's area of mangroves in 1996.

**Environmental governance.** We assessed the Environmental Performance Index (EPI)[88] as an indicator of a country's effectiveness in environmental governance. The biodiversity and habitat (BDH) issue category assesses countries' actions toward retaining natural ecosystems and protecting the full range of biodiversity within their borders. We took the BDH score for 2020 for the 2007–2016 time period and the BDH score for 2010 for the 1996–2007 time period (calculated by subtracting the ten-year change from BDH 2020). However, due to collinearity with other variables this index was excluded from the analysis (see statistical analysis).

**Protected area management.** We also assessed Marine Protected Area (MPA) staff capacity as an indicator of the effectiveness of management of protected areas for countries. We used published global marine protected area (MPA) management data[14] which is based on the Management Effectiveness Tracking Tool (METT), the World Bank MPA Score Card, and the NOAA Coral Reef Conservation Programme's MPA Management Assessment Checklist. Adequate staff capacity was the most important factor in explaining fish responses to MPA management globally, followed by budget capacity, but they were significantly correlated[14]. We, therefore, calculated the mean staff capacity across MPAs per country as our indicator. Mangroves can be included in terrestrial protected areas, which are not represented in this dataset, however, this measure provides an indicator of national governance of protected areas. However, due to collinearity with other variables this indicator was excluded from the analysis (see statistical analysis). The extent of protected areas was not included in the analysis because it has already been found to influence mangrove loss[18].

**Biophysical variables (Supplementary Table 5)**

**Coastal geomorphic type.** Mangrove extent change likely varies among different coastal geomorphic settings because human activities or environmental changes occur more commonly in some geomorphic settings than others. For example, losses of lagoonal mangroves were nearly twice as large as those in other geomorphic types[24]. Landscape geomorphic units from the global mangrove typology dataset v2.2[64] were classified as delta, estuary, lagoon or open coast.

**Sediment availability.** Mangrove expansion and retreat are driven by sediment deposition and erosion, which are influenced by sediment availability from rivers and wave action, and alterations in hydrodynamic regimes[47,89]. We used the sediment trapping index from the global free-flowing rivers (FFR) dataset[90] to indicate sediment availability from rivers within different geomorphic units. A mangrove catchment dataset was created based on the HydroSHEDS database[91]. River networks that intersected with mangrove geomorphic units were linked to that unit's ID. Where rivers intersected multiple units, they were manually assigned by visual inspection. River basins that intersected either with the geomorphic units directly or the river networks were also linked to that unit's ID. The FFR dataset[90] was then spatially joined to the mangrove catchment dataset to identify the most downstream (i.e., the coastal outlet) segment of each FFR and its associated sediment trapping index. Not all geomorphic units ($n = 3475$) were linked to an FFR, however, an individual unit could be linked with several FFRs. Therefore, the unit sediment trapping index was the weighted mean of the river values, with weighting based on each FFR's average long-term (1971–2000) naturalised discharge ($m^3s^{-1}$), with discharge set to the minimum value for segments with zero flow. Geomorphic units without connecting FFRs were given an index of zero (no sediment trapping). The sediment trapping index represents the percentage of the potential sediment load trapped by anthropogenic barriers along the river section. The focus on river barriers may obscure larger scale oceanic patterns that influence mangrove losses and gains (e.g., movement of mud banks from the Amazon River over 1000's of kilometres[92]) or increases in sediment that could be coming from soils with catchment deforestation and erosion.

**Habitat fragmentation.** Many countries with high mangrove loss have been associated with elevated fragmentation of mangrove forests, although the relationship is not consistent at the global scale[93]. We calculated the clumpiness index of mangrove patches within geomorphic units within each time period, as this habitat fragmentation metric is independent of areal extent[93]. Whilst habitat fragmentation can be human-driven, clumpiness measures the patchy distribution of mangroves, which can also be due to natural factors inducing edge effects. We used a similar approach to Bryan-Brown, et al.[86] to quantify the clumpiness index. The 'landscape' was defined as the combined extent of the mangrove geomorphic units across four timesteps (1996, 2007, 2010, and 2016) from the GMW dataset[56]. For the three focal years in this study (1996, 2007, and 2016) each geomorphic unit ($n = 4394$) was converted into a two-class polygon, where class one represented mangroves present during that time step and class two mangroves present in the other time steps (i.e., areas of mangrove loss). The polygons were transformed to a projected coordinate system (World Cylindrical Equal Area) and converted to rasters with a resolution of 25 m. Each raster was imported into R version 3.6.3[94], with clumpiness calculated using the package 'landscapemetrics' v1.5.0[95].

Clumpiness describes how patches are dispersed across the landscape and ranges between minus one, where patches are maximally disaggregated, to one, where patches are maximally aggregated, a value of zero represents a case whereby patches are randomly distributed across the landscape. The clumpiness index requires that both classes are present in the landscape, therefore a no data value

(NA) was returned for units where no loss of mangroves had occurred, or where there was 100% gain of mangroves in a later time period. The number of directions in which patches were connected was set to eight. The following manual fixes were conducted for NA values returned: 1) Where NA was returned for units where no loss of mangroves had occurred in another time period, i.e., class 1 (mangrove present) = 1 and class 2 (mangrove loss) = 0, assume +1 (maximally clumped); and 2) Where NA was returned for units where there was 100% gain of mangroves in a later time period, i.e., class 1 (mangrove present) = 0, class 2 (mangrove present) = 1 (100% gain), assume −1 (maximally disaggregated).

**Tidal amplitude.** In settings of low tidal range, mangrove vertical accretion is less likely to keep pace with rapid sea level rise[3]. However, in settings of high tidal range, mangroves may be more extensive and vulnerable to conversion to aquaculture or agriculture because of larger tidal flat extents. The Finite Element Solution global tide model (FES2014)[96] is considered one of the most accurate tide models for shallow coastal areas[97] and was selected to estimate the mean tidal amplitude within each geomorphic unit using the principal lunar semidiurnal or $M_2$ tidal amplitude as this is this most dominant tidal constituent[98]. To account for potential variation in the tidal amplitude across large geomorphic units, the raster pixel value for $M_2$ tidal amplitude[96] closest to the centroid of each mangrove patch within each unit was calculated, with the smallest value set at 0.01 m. For each geomorphic unit, the tidal amplitude was calculated as the weighted mean of the patch values, with weighting based on the patch area relative to the total unit area.

**Antecedent sea-level rise.** The distribution of mangroves on shorelines changes over time with sediment accretion, erosion, subsidence, and sea-level rise (SLR)[99], and periods of low sea level can cause mangrove dieback[100]. We used regional mean sea-level trends between January 1993 and December 2015 from the global sea level Essential Climate Variable (ECV) product v.2[101,102] to estimate the mean antecedent SLR for each geomorphic unit. Spatial variation in regional sea-level trends generally range between −5 and +5 mm yr⁻¹ (global mean of 3 mm yr⁻¹)[13]. Extreme values (>5 mm yr⁻¹) observed in the dataset are subject to high levels of uncertainty (Sea Level CCI team, pers. comm.), and were therefore truncated to 5 mm yr⁻¹. The raster pixel value for SLR[102] closest to the centroid of each mangrove patch within each geomorphic unit was calculated. The geomorphic unit antecedent SLR values was calculated as the weighted mean of the patch values within the unit.

**Drought.** Whilst long-term precipitation and temperature influence mangrove distribution globally[62], periods of low rainfall have been reported to cause extensive mangrove dieback at regional scales, particularly when combined with high temperatures and low sea levels[103]. We used the Standardized Precipitation-Evapotranspiration Index (SPEI) from the global SPEI database v.2.6[104] as an index of drought severity. SPEI is derived from precipitation and temperature and is considered an improved drought index that allows spatial and temporal comparability[105,106]. The mean SPEI raster pixel value was calculated for each time period and then averaged across the geomorphic units using the 'ncdf4'[107] and 'raster' packages[74] in R.

**Tropical storm frequency.** Large-scale destruction of mangroves across regions have been reported from strong winds, high energy waves, and storm surges associated with tropical storms[108]. We used the International Best Track Archive for Climate Stewardship (IBTrACS) dataset since 1980 v4[109] to calculate the number of tropical cyclone occurrences (points along their paths) within a 200 km buffer of the centroid of geomorphic units within each time period using the sf package[110] in R. Maximum wind velocity and surface pressures are

likely experienced within 100 km of a cyclone's eye[111], therefore the 200 km buffer zone was selected to cover the average size of geomorphic units (33.63 ha), and all tropical storms potentially influencing mangrove growth. Whilst tropical storms affect only 42% of the world's mangroves[60], they are likely to be important stressors within cyclone-impacted countries.

**Minimum temperature.** Extreme low temperature events were a driver of mangrove loss in subtropical regions, such as Florida and Louisianan of the US, and China[28,112]. We used the WorldClim bioclimatic variable 6 (minimum temperature of the coldest month averaged for the years 1970–2000)[113] to calculate the mean minimum temperature across the geomorphic units using the 'sf'[110] and 'raster' packages[74] in R. Where NAs were returned due to no overlapping raster layer, the value of the closest raster pixel to the centroid of the geomorphic unit was assigned.

**Statistical analysis.** We used multi-level linear modelling to investigate relationships between mangrove cover change variables and socioeconomic and biophysical variables to consider landscape (level 1) and country (level 2) predictors in a hierarchical approach[114]. For each response variable, we modelled the response for 1996–2007 and 2007–2016, using explanatory variables specific to the time-period where available. Data inspection revealed that high percent loss or gain was concentrated in small geomorphic units, therefore to avoid bias in our results, we removed geomorphic units less than 100 ha from the analysis, which further reduced the available sample size to 3134 units across 95 countries. Statistical analysis was undertaken in R 4.0.2[68].

The response variables were log-transformed to fit normal distribution. We tested for collinearity between our explanatory variables using Pearson's correlation coefficient ($r > 0.5$) (Supplementary Tables 6 and 7). MPA staff capacity and EPI were excluded from our models because MPA staff capacity was correlated with ECI 2007 and ECI 2016 (both $r = 0.54$), and EPI 2020 was correlated with VDEM 2016 ($r = 0.63$). To improve model fit, travel time to the nearest city, mangrove restoration effort and Ramsar wetland area (relative) were log+1-transformed, and tidal amplitude was log-transformed.

Two linear multi-level (mixed-effects) models were fitted for each response variable using the lme function in the 'lme4' package[115] (Supplementary Table 8). First, a random intercept model with intercepts of landscape-level predictors varying by country was fitted. Then a random intercept and slope (coefficients) model with intercepts of landscape-level predictors varying by country, as well as slopes for socioeconomic predictors considered to have between-country variation (travel time to nearest city and night-time lights growth) was fitted, as we expect that mangrove cover change may respond to economic growth and market accessibility depending on national governance. A likelihood ratio test between the null linear model and the null random intercept model for each response variable showed that effects varied across countries and therefore we included country as a random effect (Supplementary Table 9). We also conducted likelihood ratio tests between the random intercept model and the random coefficient model to test whether the effect of travel time and night-time lights on mangrove change varies across countries. If significant, the model including random slopes for travel time and night-time lights was used (Supplementary Table 9). Mixed-effects models were fitted by maximum likelihood and model fit was validated by inspection of residual plots for the four response variables included in the analysis; percent net loss, percent net gain, percent gross loss, and percent gross gain (Supplementary Table 9).

To test for spatial autocorrelation we performed spatial autoregressive (SAR) models using the errorsarlm function in the 'spatialreg' package[116]. SAR models were first fitted using a range of neighbourhood distances (50, 500, and 1000 km in 100 km intervals) for the net change variable[117]. Distance of 500 km showed the smallest

AIC and was therefore adopted for all response variables. Neighbourhood lists of the centroid coordinates of the geomorphic units were defined with the row-standardised ('W') coding using the 'spdep' package[118]. We then produced Moran's I correlograms using the correlog function in the 'ncf' package[119] and the centroid coordinates of the geomorphic units. Correlograms for the multi-level model and SAR model were compared for each response variable (Supplementary Fig. 4). The SAR models did not improve spatial autocorrelation for any of the mangrove cover change variables and therefore the multi-level models were adopted.

**Hotspot estimates.** We defined hotspots as geomorphic units where raw values of percent net and gross loss and gain between 2007 and 2016 ($\gamma$) differed by more than two standard deviations ($sd$) from the country average ($\mu$).

$$\text{More loss} / \text{more gain} = (\gamma - \mu) > (2 \times \text{sd}) \qquad (1)$$

$$\text{Less loss} / \text{less gain} = (\gamma - \mu) < -(2 \times \text{sd}) \qquad (2)$$

We excluded countries with only one geomorphic unit. Large deviations of the raw value from the country average were found for small units at a threshold below 50 km², therefore we removed all units smaller than 50 km² to overcome bias of hotspots towards smaller sites. This likely removed the identification of several hotspots. For example, Myanmar has had some large gains due to river sediments in the Gulf of Martaban (net gain of 100 % in Estuary 5834 and 39 % in Open Coast 62244), however, these areas were small (8 and 2 km², respectively) and were therefore removed from the hotspot estimates.

We analysed the factors contributing to hotspots by spatial investigation of satellite imagery in Google Earth with mangrove specialists from those countries. The hotspots were also assessed against protected area datasets for those countries[120–123].

**Reporting summary**

Further information on research design is available in the Nature Research Reporting Summary linked to this article.

## Data availability

The data generated in this study, including all variables for the analysis of drivers and hotspots of mangrove loss and gain, and national datasets on community forestry, NDCs, and mangrove restoration effort have been deposited in Figshare: https://doi.org/10.6084/m9.figshare.21097435. The number of sampling units per country used in the analysis and the national extent of mangrove cover in 1996, 2007, and 2016 are provided in Supplementary Data 1. Raw data used in the study for the calculation of variables are available as follows: Global Mangrove Watch 1996-2016 dataset from the UNEP World Conservation Monitoring Centre (WCMC) (http://data.unep-wcmc.org/datasets/45)[56], global biophysical mangrove typology dataset from the UNEP WCMC (https://data.unep-wcmc.org/datasets/48)[64], country data from GADM (https://gadm.org/data.html)[65], Exclusive Economic Zones in the Maritime Boundaries Geodatabase from VLIZ (https://doi.org/10.14284/386)[66], Night-time Lights Time Series from the National Geophysical Data Centre (https://ngdc.noaa.gov/eog/dmsp/downloadV4composites.html)[72], Economic Complexity Index from the Observatory of Economic Complexity (https://oec.world/en/rankings/eci/hs4/hs92)[76], VDEM index from Varieties of Democracy (https://doi.org/10.23696/vdemds20)[77], community forestry projects from the Global ICCA Registry (https://www.iccaregistry.org/)[82] and the International Database on REDD + projects and programmes (http://www.reddprojectsdatabase.org)[83], National Determined Contributions from the NDC registry (https://www4.unfccc.int/sites/ndcstaging/Pages/Home.aspx)[85], Ramsar sites from the Ramsar Sites Information Service (https://rsis.ramsar.org/)[87], Environmental Performance Index from the Yale Centre (https://epi.yale.edu/downloads)[88], marine protected area management data in Supplementary Data 1 from Gill, et al.[14], Sediment Trapping Index in the global free-flowing rivers dataset from figshare (https://doi.org/10.6084/m9.figshare.7688801)[124], river networks from the HydroSHEDS database (https://www.hydrosheds.org/hydrosheds-core-downloads)[125], tidal amplitude in FES2014 from the Aviso+ Cnes Data Centre (https://www.aviso.altimetry.fr/)[96], sea-level rise in the global sea level ECV product from the CCI Open Data Portal (https://doi.org/10.5270/esa-sea_level_cci-MSLA-1993_2015-v_2.0-201612)[102], SPEI from Digital CSIC (https://digital.csic.es/handle/10261/202305)[104], tropical storms in IBTrACS from the NOAA National Centres for Environmental Information (https://doi.org/10.25921/82ty-9e16)[109], and bioclimatic variables from WorldClim (https://www.worldclim.org/data/worldclim21.html)[113]. Databases used were publicly available and permission was not required.

## Code availability

R code developed for the driver and hotspot analysis, figure production, and for the calculation of community forestry, NDC and mangrove restoration indices are available in Figshare: https://doi.org/10.6084/m9.figshare.21097435. R code can be used together with the published datasets (see "Data availability") to recalculate the study results.

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

## Acknowledgements
This study was funded by an Australian Research Council linkage grant (LP170101171) with financial support from The Nature Conservancy and Healthy Land and Water. M.I.S. was supported by a Julius Career Award from CSIRO and acknowledges the Coasts and Ocean Programme 'Nature-Based Solutions and Restoration' research domain at CSIRO Oceans and Atmosphere. T.W. was supported by an anonymous gift to The Nature Conservancy. C.L. was supported by an Australian Laureate fellowship (FL200100133).

## Author contributions
M.S., C.L., E.L., and V.H. conceived the idea for this study. All authors participated in a collaborative workshop, contributed to the conceptualisation of hypotheses and indicators, and edited the manuscript. V.H. and T.W. calculated the mangrove change, socioeconomic and biophysical indicators. T.A. assisted with the modelling and statistical analysis. V.H. undertook the analysis and prepared the manuscript.

## Competing interests
The authors declare no competing interests.
