## [Peer Review File · Nature Communications]

Reviewers' Comments:

Reviewer #1:

Remarks to the Author:

This ms addressed a very important and interesting issue. It linked the social economy with ecological systems very well. Thus, the key scientific question on mangrove area changes were fitting the changes of regional economic growth. Furthermore, this study also reflected the correlation between ecological functions and the assess to markets in the global scale, which fit the gaps and combined the knowledge of ecological restoration and conservation with the social development. That is very innovative.

For the methods, this ms constructed a large number of datasets from the previous study, reports and remote sensing, which were classified. The change of mangrove area was divided into two decades of periods, 1996-2007 and 2007-2016, which was very smart because the time scale reflecting the different point of view of mangrove ecosystem linking to NDC and blue carbon. To be mention, the second decade was right in the fast development time of people's awareness of blue carbon. The most obvious changes between the two decades is the efforts to climate mitigating actions. Therefore, this ms would be constructive.

Here, I have some comments to the authors for the further consideration in the revisions.

1. In the driving forces of the area change in mangroves worldwide, the authors took into account of climate extremes and lined out storms and drought. I would suggest to include the extreme low temperature events that is very serious in mangroves in subtropical regions. Some previous studies attributed this driver to mangroves loss in Florida and Louisianan of the US, mangroves in China and also other regions (Ross et al., 2009; Osland et al., 2014; Chen et al., 2017, etc.). The intensity of extreme low temperature would be increased in the future warming scenarios. Those mangroves located in the northern limits or southern limits may face this stress. Including this factor would be more rigorous.
2. The Extended data figure 2 is very informative, maybe it would be good to move to the major part of results.
3. The authors analyzed mangrove area increasing drivers. Among them, the sediment accretion was very important in Asia. I agree and suggest to consider the combined effects of restoration or afforestation with accretion, since there would be a positive feedback of forest management, such as afforesting or thinning, especially the actions in the seaward edges of mangrove forests that has intensely expand mangrove areas. This situation is also common in China. I suggest to include the case from China in the Extended data table 2.

Reviewer #2:

Remarks to the Author:

The manuscript is a timely one in terms of contemporary global environmental changes and deals with the topic efficiently in most of the sections. I like to suggest few revisions/ modifications to make the quality and appeal of the manuscript considerably better:

- 1.The 'abstract' mentions of the 'landscape-scale' analysis of mangrove changes, but it has to be clarified what authors wanted to mean by landscape, is it the landscape in its true sense as per the understandings of landscape ecology encompassing both mangrove and non-mangrove patches or the administratively delineated boundary of mangrove forests that regulates the actual management regimes?
- 2.The authors highlighted the importance of co-management and community based management of mangroves, yet the list of references do not adequately covers some of the most relevant and highly cited articles in this context. The reference section has to be upgraded to address this issue.
- 3.In line 62-64, authors have discussed on the paucity of information on the effects of community forestry in the management of mangroves throughout the world. This claim seems to be too generalized as there are several regional/ location-specific precise accounts of such effects on mangrove forests. Of course, there are also data-poor areas in many parts. Thus, it would be better to term this gap as a regional one instead of global.
- 4.hat is the basis of selecting 1996 as the initial year of study? Is it only due to the availability of the GMW dataset? This is important since the choice of decadal period depends on it and thereby affects the statistical inferences.
5. Many times, relationships have been mentioned as 'strong correlation' without clarifying the

positive or negative trend of those. This aspect has to be revised.

6. In line 160, NDC commitments had been analyzed at the national scale, but it is unclear whether the gain or loss estimates were in national or landscape scale.

7. Figure 1d and 1e should be of better quality.

8. Please differentiate 'positive loss' and 'negative gain'. If same, then can these be merged into a single figure for comprehensive representation instead of Fig. 1 and 2.

9. Why Fig 1C is represented on map and not in scatter plots like those of Fig. 2 c, d, e and vice versa?

10. Fig. 3 has a serious mistake where the Indian geographical territory has been labelled as PAK, probably due to cartographic adjustments during the naming of a mangrove loss area in Pakistan. This is unacceptable in such a high quality journal.

11. Few countries, such as India, Iran, Kenya and Tanzania, having large to moderate mangrove covers as well as Ramsar/ World Heritage Sites within their mangrove territories had not been addressed properly either in the tables/ figures or in the main text. Slight modification of the text will help the manuscript truly global.

REVIEWER COMMENTS

Reviewer #1 (Remarks to the Author):

This ms addressed a very important and interesting issue. It linked the social economy with ecological systems very well. Thus, the key scientific question on mangrove area changes were fitting the changes of regional economic growth. Furthermore, this study also reflected the correlation between ecological functions and the assess to markets in the global scale, which fit the gaps and combined the knowledge of ecological restoration and conservation with the social development. That is very innovative.

For the methods, this ms constructed a large number of datasets from the previous study, reports and remote sensing, which were classified. The change of mangrove area was divided into two decades of periods, 1996-2007 and 2007-2016, which was very smart because the time scale reflecting the different point of view of mangrove ecosystem linking to NDC and blue carbon. To be mention, the second decade was right in the fast development time of people's awareness of blue carbon. The most obvious changes between the two decades is the efforts to climate mitigating actions. Therefore, this ms would be constructive.

Here, I have some comments to the authors for the further consideration in the revisions.

1. In the driving forces of the area change in mangroves worldwide, the authors took into account of climate extremes and lined out storms and drought. I would suggest to include the extreme low temperature events that is very serious in mangroves in subtropical regions. Some previous studies attributed this driver to mangroves loss in Florida and Louisianan of the US, mangroves in China and also other regions (Ross et al., 2009; Osland et al., 2014; Chen et al., 2017, etc.). The intensity of extreme low temperature would be increased in the future warming scenarios. Those mangroves located in the northern limits or southern limits may face this stress. Including this factor would be more rigorous.

We originally considered including maximum temperature changes in the biophysical variables, however excluded it because of relevance mainly to mangrove distributional limits. However, we agree that your suggestion of including extreme low temperatures may be relevant to mangrove dieback in subtropical regions (Ross et al. 2019, Chen et al. 2017) as well as expansion in northern and southern range limits (Osland et al. 2014). We have therefore included minimum temperature of the coldest month (bioclimatic variable 6) from WorldClim as a metric of extreme low temperature.

The WorldClim dataset was analysed to obtain mean minimum temperature for each mangrove geomorphic unit. This new variable was incorporated into the models for each mangrove change indicator and the statistical analysis redone. Methods, results, figures, and supplementary material have been updated accordingly. We found that there was a significant negative relationship between mangrove losses and minimum temperature in the first decade, demonstrating higher losses with lower minimum temperatures. There was also a significant negative relationship between mangrove gains and minimum temperature, demonstrating regeneration in areas with lower minimum temperatures in the first decade. As a result of including minimum temperature, the significant effects of the other extreme weather variables on mangrove net gain were removed – drought in the first decade and tropical cyclones in the second decade. Furthermore, travel time to nearest city was no longer negatively significant with mangrove net gain in the second decade, and mangrove restoration effort emerged as positively significant with mangrove net gain in the first decade. However, these changes do not affect the implication of the results.

We have updated the results in the main text:

“While there was evidence of mangrove dieback in response to severe droughts¹⁴ and extreme low temperatures³⁰(Ross et al. 2009) in the first decade, demonstrated as higher loss with lower SPEI and lower minimum temperature of the coldest month (net loss %, Fig. 2a; gross loss %, Extended data fig. 5i), mangrove gains were also higher in areas with lower minimum temperatures in both decades, indicating potential regeneration after disturbance⁵² (net gain %, Fig. 3a; gross gain %, Extended data fig. 5g,h). (lines 213-219)

We have added into the methods:

“Minimum temperature: Extreme low temperature events were a driver of mangrove loss in subtropical regions, such as Florida and Louisiana of the US, and China^{28,102}(Ross et al. 2009, Chen et al. 2017). We used the WorldClim bioclimatic variable 6 (minimum temperature of the coldest month averaged for the years 1970-2000)¹⁰³(Fick et al. 2017) to calculate the mean minimum temperature across the geomorphic units using the sf and raster packages in R. Where NAs were returned due to no overlapping raster layer, the value of the closest raster pixel to the centroid of the geomorphic unit was assigned.”

Additional references cited in the main text, methods and supplementary information:

Ross, M. S., Ruiz, P.L., Sah, J.P., Hanan, E.J. (2009). Chilling damage in a changing climate in coastal landscapes of the subtropical zone: a case study from south Florida. *Global Change Biology*, 15, 1817–1832.

Chen, L., Wang, W., Li, Q. Q., Zhang, Y., Yang, S., Osland, M. J., Huang, J., & Peng, C. (2017). Mangrove species' responses to winter air temperature extremes in China. *Ecosphere*, 8(6), e01865.

Fick, S. E., & Hijmans, R. J. (2017). WorldClim 2: new 1km spatial resolution climate surfaces for global land areas. *International Journal of Climatology*, 37 (12), 4302-4315.
<https://www.worldclim.org/data/monthlywth.html>

Osland, M. J., Day, R. H., Larriviere, J. C., & A.S., F. (2014). Aboveground Allometric Models for Freeze-Affected Black Mangroves (*Avicennia germinans*): Equations for a Climate Sensitive Mangrove-Marsh Ecotone. *PLOS ONE*, 9(6), e99604. (In the supplementary material)

2. The Extended data figure 2 is very informative, maybe it would be good to move to the major part of results.

We have moved this to the main results as a new Fig. 1.

3. The authors analyzed mangrove area increasing drivers. Among them, the sediment accretion was very important in Asia. I agree and suggest to consider the combined effects of restoration or afforestation with accretion, since there would be a positive feedback of forest management, such as afforesting or thinning, especially the actions in the seaward edges of mangrove forests that has intensely expand mangrove areas. This situation is also common in China. I suggest to include the case from China in the Extended data table 2.

In selecting our biophysical variables, we recognised that mangrove loss and gain are driven by sediment accretion and erosion, which are in turn influenced by sediment availability from rivers and wave action, and alterations in hydrological regimes (Thomas et al. 2017, Twilley et al. 2017, van Bijsterveldt et al. 2020 - see Table S3 in the supplementary information). At the global scale there is no data available on sediment accretion. Therefore, we choose the sediment trapping index from the global free-flowing rivers dataset as our metric, as sediment retention because of dam

construction in rivers has been found to contribute significantly to mangrove decline on a global-scale (Maynard et al. 2019, Syvitzki et al. 2009). We found the sediment trapping index to be a driver of mangrove loss (% gross loss), but not of mangrove gain. Increased loss was associated with reduced sediment availability, likely due to lower levels of sediment accretion that lead to the dominance of erosive forces on minerogenic mangrove shorelines (see line 44). The non-significant effect of sediment trapping index on mangrove gain is likely because of limitations with the sediment trapping index in representing sediment supply from oceanic sources (only a metric of fluvial sediment supply). Furthermore, your comment on the combined effects of restoration or afforestation with accretion is important. We acknowledge the possibility for an interaction between mangrove restoration/afforestation and sediment accretion on mangrove gain, however we haven't been able to include it in the analysis due to the absence of data on sediment supplies at the global scale.

We have added into the main text:

“However, sediment availability was not associated with mangrove gain, likely because the sediment trapping index did not measure longshore sediment supplies or sediment increases that could be coming from soils with catchment deforestation and erosion. Seaward expansion of mangrove forests has been associated with tidal flat accretion from sediment transported by tidal currents and waves, offsetting large declines of fluvial sediment supply (Long et al. 2022) and further facilitating sediment capture (Chow 2018). Thus, there is a data limitation at the global scale to assess sediment processes on mangrove change.” (lines 204-210)

Additional reference cited:

Long, C., Dai, Z., Wang, R., Lou, Y., Zhou, X., Li, S., & Nie, Y. (2022). Dynamic changes in mangroves of the largest delta in northern Beibu Gulf, China: Reasons and causes. *Forest Ecology and Management*, 504, 119855.

Chow, J. (2018). Mangrove management for climate change adaptation and sustainable development in coastal zones. *Journal of Sustainable Forestry*, 37(2), 139-156.

Reviewer #2 (Remarks to the Author):

The manuscript is a timely one in terms of contemporary global environmental changes and deals with the topic efficiently in most of the sections. I like to suggest few revisions/ modifications to make the quality and appeal of the manuscript considerably better:

1.The 'abstract' mentions of the 'landscape-scale' analysis of mangrove changes, but it has to be clarified what authors wanted to mean by landscape, is it the landscape in its true sense as per the understandings of landscape ecology encompassing both mangrove and non-mangrove patches or the administratively delineated boundary of mangrove forests that regulates the actual management regimes?

Here we define landscape-scale as the delineation of mangrove cover based on geomorphic settings into mangrove biophysical typologies (delta, estuary, open coast, lagoon) (Worthington et al. 2020). We used the composite model from Worthington et al. 2020, which is the maximal extent from the combined Global Mangrove Watch 1996, 2007, 2010, and 2016 timesteps. It does not include non-mangrove patches, unless they have been lost from the typology over the timesteps.

We have clarified this in the main text and in the methods under mangrove cover change variables.

"Mangrove geomorphic units were delineated by the maximal extent of mangrove cover across 1996-2016 and classified into typologies based on geomorphic settings (see Methods)." (lines 76-78)

"A global mangrove geomorphic typology dataset v2.2 was used for the delineation of landscape mangrove geomorphic units (Worthington et al. 2020), which used a composite of the GMW dataset from the 1996, 2007, 2010, and 2016 timesteps to classify the maximal extent of mangrove cover into 4394 units (classified as delta, estuarine, lagoon or open coast). The mangrove geomorphic units do not include non-mangrove patches, unless they have been lost from the unit over time." (Methods)

2.The authors highlighted the importance of co-management and community based management of mangroves, yet the list of references do not adequately covers some of the most relevant and highly cited articles in this context. The reference section has to be upgraded to address this issue.

We have upgraded the discussion on community-based management of mangroves to include four more key references that were considered in the systematic literature review to generate the dataset on national community forestry effort – two global and two mangrove specific.

"Community forest management or co-management can lead to positive conservation and social outcomes globally particularly in communities with de facto tenure rights and countries with low development and governance indicators (Hajjar et al. 2021). For instance, titling indigenous communities has been shown to decrease clearing and disturbance in the Peruvian Amazon forests (Blackman et al. 2017). Increased national commitment to community forestry policies and programs could potentially reverse mangrove losses in other countries, such as Kenya and Myanmar (Frank et al. 2017, Feurer et al. 2018), however, there are issues with the governance and tenure of mangroves at the land-sea interface that need resolving⁴¹." (lines 162-169)

Additional references cited:

Hajjar, R. et al. (2021) A global analysis of the social and environmental outcomes of community forests. *Nat. Sustain.* 4, 216-224.

Blackman, A., Corral, L., Lima, E. S. & Asner, G. P. (2017) Titling indigenous communities protects forests in the Peruvian Amazon. *Proc. Natl. Acad. Sci. USA*, 114, 4123-4128.

Frank, C., Kairo, J. G., Bosire, J. O., Mohamed, M. O. S., Dahdouh-Guebas, F., & Koedam, N. (2017). Involvement, knowledge and perception in a natural reserve under participatory management: Mida Creek, Kenya. *Ocean & Coastal Management*, 142, 28-36.

Feurer, M., Gritten, D., & Than, M. M. (2018). Community Forestry for Livelihoods: Benefiting from Myanmar's Mangroves. *Forests*, 9(3), Article 150.

3. In line 62-64, authors have discussed on the paucity of information on the effects of community forestry in the management of mangroves throughout the world. This claim seems to be too generalized as there are several regional/ location-specific precise accounts of such effects on mangrove forests. Of course, there are also data-poor areas in many parts. Thus, it would be better to term this gap as a regional one instead of global.

We agree that there are regional accounts of the effects of community forestry on mangroves and we mention some of these studies in the discussion on community forestry (Camacho et al. 2020 and Sudtongkong et al. 2007) and have added two more as per previous comment. We have amended the statement so that it is not so generalised:

“In Myanmar and Kenya, community forestry is being encouraged in mangroves to improve livelihoods and mangrove conservation (Frank et al. 2017, Feurer et al. 2018), however the effects of such initiatives on mangrove forests on a global-scale are unknown and represent a major gap in our understanding of how we can reverse mangrove losses;” (lines 63-67)

Camacho, L., Gevaña, D., Sabino, L., Ruzol, C., Garcia, J., Camacho, A., Oo, T., Maung, A., Saxena, K., Liang, L., Yiu, E. and Takeuchi, K. Sustainable mangrove rehabilitation: Lessons and insights from community-based management in the Philippines and Myanmar. *APN Science Bulletin* 10, (2020).

Sudtongkong, C. & Webb, E. L. Outcomes of state- vs. community-based mangrove management in southern Thailand. *Ecology and Society* 12, 27 (2008).

4. What is the basis of selecting 1996 as the initial year of study? Is it only due to the availability of the GMW dataset? This is important since the choice of decadal period depends on it and thereby affects the statistical inferences.

It is based on the availability of mangrove cover data, in this case the GMW dataset and available timesteps (1996, 2007, 2008, 2009, 2010, 2015, 2016). We have clarified in the introduction (changes in bold):

“...assessing two decades allowed us to account for recent reductions in mangrove loss²⁵, **within the available timesteps of the GMW dataset.**” (lines 75-76)

5. Many times, relationships have been mentioned as ‘strong correlation’ without clarifying the positive or negative trend of those. This aspect has to be revised.

We have followed the reviewer’s suggestion and have clarified the directionality of the relationships. This requires some contextualisation in the text, because some of the indices had opposite ‘directions’ compared to their impact. For instance, low clumpiness = high fragmentation, and low Standardized Precipitation-Evapotranspiration Index (SPEI) = high drought severity. Therefore, net loss was negatively correlated with clumpiness and SPEI, however this means that there was higher loss with higher fragmentation and higher drought severity.

In paragraph 5, where we summarise the results of the models, I have added the trends of the associations. However, in the discussion, to avoid confusion, I have clarified the associations as follows (changes are in bold):

“As expected, **higher** mangrove fragmentation (**lower clumpiness**) was strongly associated with greater mangrove loss.” (lines 198-199)

“Increased loss was also associated with reduced sediment availability (**higher sediment trapping**),” (lines 201-202)

“While there was evidence of mangrove dieback in response to severe droughts¹⁴ **and extreme low temperatures**⁵¹ in the first decade, **demonstrated as higher loss with lower SPEI and lower minimum temperature of the coldest month**,” (lines 213-215)

“...mangrove gains were also **higher** in areas with **lower minimum temperatures** in the first decade,” (lines 216-217)

6. In line 160, NDC commitments had been analyzed at the national scale, but it is unclear whether the gain or loss estimates were in national or landscape scale.

The explanatory variable (NDC commitments) was at the national scale, however the response variable (loss or gain) was at the landscape scale. It was modelled in a multi-level model to look at both individual- and group-level effects on mangrove loss or gain. We have added to clarify:

“...there was no association between **national** NDC commitment or restoration effort with reducing rates of loss or increasing rates of gain **across mangrove geomorphic units** at the end of the analysis period (2017).” (lines 177-179)

7. Figure 1d and 1e should be of better quality.

We have revised to combine the time periods into one plot so that you can clearly see the difference across the time periods (revised Fig. 2f). We have undertaken this revision for each response variable to be consistent (revised Fig. 3f, Extended data fig. 2f and 4f).

8. Please differentiate ‘positive loss’ and ‘negative gain’. If same, then can these be merged into a single figure for comprehensive representation instead of Fig. 1 and 2.

Thank you for pointing out this lack of clarity in our manuscript. Here we provide some context and we have made annotations to the text to provide clarity to the reader. We like the idea of combining the loss and gain variables, but we cannot do so for the following reasons:

Net loss and gain were spatially distinct response variables. Firstly, mangrove cover net change was calculated for each mangrove geomorphic unit, ranging from negative (loss) to zero (no change) to positive (gain). Initially, we explored using the net change response variable in the modelling as no change is an important conservation outcome. Net change showed high frequency around zero and therefore there were problems with model fit (see Table S.6) and we were unable to adopt this model (see Supplementary S.10). Therefore, it was necessary to separate loss and gain and convert loss into a positive number in order to log the response variables to achieve normal distribution. In this analysis, if net change was positive, it was assigned to the gain variable. If net change was negative, it was assigned to the loss variable. If there was no net change, that geomorphic unit was omitted from the dataset.

Although decreased loss and increased gain are both conservation outcomes, they have been modelled separately, with the figures being displayed separately to show the results of the models.

I have added to the methods to explain this logic (changes in bold): “We initially considered 10 different response variables (Table S1), **including net change in mangrove cover ranging from**

negative (loss) to zero (no change) to positive (gain), however the data did not meet the model assumptions and it was necessary to separate net loss and gain (method in S.1)." (Methods)

I have also added some more details to Supplementary S.1 (changes in bold): "Although no change in mangrove cover (0) is a positive conservation outcome, we cannot analyse no change because **the data did not meet model assumptions of normal distribution and equal variance** (Table S.6).

Therefore, it was necessary to separate net loss and gain in order to remove zeros, convert loss to positive, and log the response variables to achieve normal distribution."

9. Why Fig 1C is represented on map and not in scatter plots like those of Fig. 2 c, d, e and vice versa?

To maintain consistency across former Fig. 1 and 2 (revised Fig. 2 and 3), we have included the relationships of travel time, nighttime lights, restoration effort/community forestry as scatterplots in both figures (net loss % and net gain %). As travel time was significant for net loss % 2007-2016, but not net gain % 2007-2016, we have moved the map of the effect of travel time across countries to revised Extended data fig. 3, with the figures showing the influence of national level governance indicators on the travel time effect on net loss % 2007-2016.

We have also maintained this consistency for the figures of gross loss % and gross gain % (revised Extended data fig. 2 and 3).

10. Fig. 3 has a serious mistake where the Indian geographical territory has been labelled as PAK, probably due to cartographic adjustments during the naming of a mangrove loss area in Pakistan. This is unacceptable in such a high quality journal.

The label is for the hotspot on gross loss % in Delta 8765, which is just within Pakistan. We have repositioned the labels to above to fix this (revised Fig. 4a).

11. Few countries, such as India, Iran, Kenya and Tanzania, having large to moderate mangrove covers as well as Ramsar/ World Heritage Sites within their mangrove territories had not been addressed properly either in the tables/ figures or in the main text. Slight modification of the text will help the manuscript truly global.

At a regional scale, lower mangrove loss rates have been reported in selected Ramsar sites in Australian, India/Bangladesh, Cameroon and Ecuador (Hamilton and Casey 2016) and we referenced this study in the main text. Viewing our data, there are a few countries that have large to moderate mangrove cover as well as moderate Ramsar extent relative to the area of mangrove cover, including Brazil, Mexico, Australia, Mozambique, Gabon, Pakistan, China and Iran. However, we found no effect on loss or gain, although Gabon had relatively low net loss in 2007-2016 and Pakistan and China experienced net gains in 2007-2016.

We have modified the main text to clarify the countries that have large to moderate Ramsar wetlands as follows. The extended data tables 1 and 2 present the results from the hotspot analysis, therefore, we cannot add additional sites.

"Counter to other studies at regional-scales **showing mangrove losses 50% lower than the global average in selected Ramsar sites**³⁰, the area of Ramsar wetlands per country was not positively associated with mangrove conservation at the national-scale. This is **despite several countries having large to moderate mangrove cover and Ramsar extent relative to their area of mangrove cover, including Brazil, Mexico, Australia, Mozambique, Gabon, Pakistan, China and Iran.** Many mangrove-holding countries are not party to the Ramsar convention and weaknesses in Ramsar governance occurs in urban wetlands in some countries⁴⁸. (lines 189-196)

Reviewers' Comments:

Reviewer #1:

Remarks to the Author:

The revised manuscript has been revised in great detail to address the comments and to respond with high clarity. The revised manuscript is now quite complete. I believe this manuscript will serve as a good guide for the future management, restoration and development of mangroves worldwide. I have no further suggestions or comments to the revised manuscript.

Reviewer #2:

Remarks to the Author:

The revisions made according to the suggestion of the Reviewers are appropriate and greatly enhanced the quality of the article. The authors have satisfactorily addressed my queries. Congratulations to them for this exceptional work!